# Pool size estimations for dense-core vesicles in mammalian CNS neurons

Claudia M Persoon[1], Alessandro Moro[1], Joris P Nassal[1], Margherita Farina[2], Jurjen H Broeke[1], Swati Arora[2] (ID), Natalia Dominguez[1], Jan RT van Weering[1], Ruud F Toonen[2,*] & Matthijs Verhage[1,2,**] (ID)

## Abstract

Neuropeptides are essential signaling molecules transported and secreted by dense-core vesicles (DCVs), but the number of DCVs available for secretion, their subcellular distribution, and release probability are unknown. Here, we quantified DCV pool sizes in three types of mammalian CNS neurons *in vitro* and *in vivo*. Super-resolution and electron microscopy reveal a total pool of 1,400–18,000 DCVs, correlating with neurite length. Excitatory hippocampal and inhibitory striatal neurons *in vitro* have a similar DCV density, and thalamo-cortical axons *in vivo* have a slightly higher density. Synapses contain on average two to three DCVs, at the periphery of synaptic vesicle clusters. DCVs distribute equally in axons and dendrites, but the vast majority (80%) of DCV fusion events occur at axons. The release probability of DCVs is 1–6%, depending on the stimulation. Thus, mammalian CNS neurons contain a large pool of DCVs of which only a small fraction can fuse, preferentially at axons.

**Keywords** dense-core vesicles; pool sizes; release probability; secretion
**Subject Categories** Neuroscience
**The EMBO Journal (2018) 37: e99672**

## Introduction

Regulated secretion of chemical signals in the nervous system occurs principally from two organelles, synaptic vesicles and dense-core vesicles (DCVs). In neurons, synaptic vesicles release neurotransmitters at synapses, whereas neuronal DCVs transport and secrete neuropeptides and neurotrophins that regulate brain development, neurogenesis, and synaptic plasticity (Park & Poo, 2013; van den Pol, 2012; Zaben & Gray, 2013; Pang *et al*, 2004; Malva *et al*, 2012). Alterations in neuropeptide levels in the brain have been associated with many pathological states, such as cognitive, anxiety, stress, and addiction related disorders (Morales-Medina *et al*, 2010; McGinty *et al*, 2010; Meyer-Lindenberg *et al*, 2011).

DCVs are filled with cargo at the *trans*-Golgi network, undergo maturation steps (Kim *et al*, 2006), and are transported throughout the neuron by microtubule-based motor proteins of the kinesin and dynein protein families (Zahn *et al*, 2004; Lo *et al*, 2011), and upon action potential-triggered calcium influx, DCVs can fuse with the plasma membrane (Gärtner & Staiger, 2002; Frischknecht *et al*, 2008; van de Bospoort *et al*, 2012; de Wit *et al*, 2009; Hartmann *et al*, 2001; Matsuda *et al*, 2009; Farina *et al*, 2015).

Dense-core vesicle fusion requires repetitive and more prolonged stimulation compared to synaptic vesicle fusion (Lundberg *et al*, 1986; Hartmann *et al*, 2001; Balkowiec & Katz, 2002; Gärtner & Staiger, 2002; Frischknecht *et al*, 2008). Synaptic vesicles in neurons and DCVs in endocrine cells have been assigned to different pools depending on their ability to undergo exocytosis. Estimations of synaptic vesicle pool sizes in cultured hippocampal neurons show a total population of 100–200 synaptic vesicles per bouton, containing a fast ready releasable pool (RRP) of approximately five vesicles that are immediately available for secretion upon stimulation, which can be resupplied by a recycling pool of 10–20 vesicles (Neher, 2015; Rizzoli & Betz, 2005; Fernández-Alfonso & Ryan, 2006). Pool sizes for neuronal DCVs have not been characterized. We have previously shown that cultured mouse hippocampal neurons and human induced pluripotent stem cell (iPSC)-derived neurons release approximately 10–100 DCVs upon repetitive stimulation (Arora *et al*, 2017; Farina *et al*, 2015; van de Bospoort *et al*, 2012; Emperador Melero *et al*, 2017). However, the number of DCVs present in mammalian neurons and probability of their release upon stimulation are currently unknown.

Dense-core vesicles are transported to axons and dendrites by KIF1 motors (Lipka *et al*, 2016), but it is unclear whether DCVs have a preference for axonal or dendritic release. Previous studies have reported fusion of brain-derived neurotrophic factor (BDNF)-containing DCVs only at dendritic (Hartmann *et al*, 2001), axonal (Dieni *et al*, 2012), or both locations (Dean *et al*, 2009; Matsuda *et al*, 2009). Furthermore, BDNF-containing DCVs have different fusion properties depending on their location of fusion (Matsuda *et al*, 2009; Dean *et al*, 2009). Incomplete cargo release was observed at axons, while full fusion events occurred at dendrites (Matsuda *et al*, 2009; Dean *et al*, 2009), indicating more quantal

1   Clinical Genetics, VU Medical Center, Amsterdam, The Netherlands
2   Department of Functional Genomics, Faculty of Exact Science, Center for Neurogenomics and Cognitive Research, VU University Amsterdam and VU Medical Center, Amsterdam, The Netherlands
    *Corresponding author. Tel: +31 0 20 59 89464; E-mail: ruud.toonen@cncr.vu.nl
    **Corresponding author. Tel: +31 0 20 59 86936; Fax: +31 0 20 598 6926; E-mail: matthijs@cncr.vu.nl

BDNF release at dendrites. However, differences in release probability of DCVs at axons or dendrites have not been studied.

In this study, we characterized the total DCV pool in mammalian neurons using immunostainings, confocal, electron, and super-resolution microscopy. The DCV pool in cultured neurons ranged from 1,400 to 18,000 DCVs per neuron and highly correlated with neuron size. The number of DCVs per μm neurite was consistent between different culture systems, but slightly increased in labeled axons in *in vivo* sections of mouse cortex. Measurements of DCV fusion events showed a releasable pool of 20–400 vesicles, corresponding to 1–6% of the total pool, depending on the type of stimulation. Furthermore, we found that approximately 80% of DCV fusion events occurred at the axon. These data provide the first characterization of total and releasable pools of DCVs in mammalian neurons as well as insights into the subcellular location of DCVs and their release.

## Results

### The total DCV number per neuron correlates with neurite length and is comparable *in vivo* and *in vitro*

To estimate the total number of DCVs per neuron, we first analyzed single neurons cultured on glia micro-islands, sampled from two brain areas, hippocampus and striatum. These neurons were immunostained for the endogenous DCV cargo chromogranin B (ChgB), a secretory granule matrix protein involved in vesicular accumulation and storage of peptides and calcium (Bartolomucci *et al*, 2011; Fig 1A and B). Quantification of vesicular glutamate transporter 1 (VGLUT1)- or vesicular GABA transporter (VGAT)-positive neurons showed that hippocampal cultures predominantly contain glutamatergic neurons (Fig 1C), while GABAergic neurons predominate in striatal cultures (Fig 1G). To estimate the total number of DCVs per neuron, ChgB puncta were retrieved from 3D confocal image stacks and counted using automated analysis

software SynD (Schmitz *et al*, 2011). Hippocampal neurons contained on average 4,000 ChgB puncta ranging from 750 to 10,000 puncta per neuron (Fig 1D). The total neurite length, quantified using β3-tubulin immunostaining, showed similar variation and was on average 7.35 mm (Fig 1E), resulting in a distribution of approximately 0.54 ChgB puncta per μm neurite (Fig 1F). Striatal neurons contained fewer DCVs, on average 1,360 ChgB puncta (Fig 1H), were smaller than hippocampal neurons (approximately 2.2 mm, Fig 1I), resulting in 0.64 ChgB puncta per μm neurite (Fig 1J). For hippocampal as well as striatal neurons, the number of ChgB puncta per neuron correlated with neurite length (Fig 1K).

To test whether these estimates of DCV density in cultured neurons are comparable to mouse neurons *in vivo*, we labeled thalamo-cortical axonal projections by stereotaxic injections in the right medio-dorsal thalamus of adeno-associated virus (AAV) expressing mCherry driven by the CAG promoter (Fig 1L). Cortical slices were stained with the canonical DCV cargo ChgA. ChgA puncta were quantified in mCherry-positive axons (Fig 1M–O), and 0.53 ChgA puncta per μm axon were detected (Fig 1P). These data show that different populations of neurons show a similar distribution of DCVs, correlating with neurite length, and similar to DCV density in axonal projections *in vivo*.

### Unitary cargo intensity and super-resolution imaging reveal total DCV pool

The total DCV pool is likely underestimated when counting ChgB puncta in confocal microscopy, due to accumulation of multiple DCVs in resolution-limited space (Fig 2A). To address this, we took two approaches: We analyzed ChgB staining intensity as a proxy for the number of DCVs per punctum and performed super-resolution fluorescence imaging by direct stochastic optical reconstruction microscopy (*d*STORM) of ChgB to correlate ChgB intensity measurements at the confocal microscopy level with the number of DCVs resolved with *d*STORM (Fig 2B–D). We assume that densely clustered reconstructed *d*STORM localizations of ChgB represent one

---

**Figure 1. Number and distribution of DCV puncta in neurons *in vitro* and *in vivo*.**

A  Representative composite confocal image of single cultured hippocampal mouse neurons, labeled with MAP2 (green) on islands of glial cells, labeled with GFAP (red). Scale bar: 100 μm.

B  Representative confocal image of single cultured hippocampal mouse neuron labeled with endogenous DCV cargo ChgB (green) and β3-tubulin (red) as neurite marker. Scale bar: 20 μm.

C–F  Analysis of hippocampal neurons (DIV14) immunostained for ChgB as DCV marker and β3-tubulin as morphological marker. (C) Percentage of glutamatergic and GABAergic neurons in hippocampal cultures determined by immunostainings for VGLUT1 and VGAT, respectively. (D) DCV pool size measured as number of ChgB puncta per neuron. (E) Total neurite length (mm) per neuron. (F) ChgB puncta per μm neurite.

G–J  Analysis of striatal neurons (DIV14) immunostained for ChgB as DCV marker and β3-tubulin as morphological marker. (G) Percentage of glutamatergic and GABAergic neurons in striatal cultures determined by immunostainings for VGLUT1 and VGAT, respectively. (H) DCV pool size measured as number of ChgB puncta per neuron. (I) Total neurite length (mm) per neuron. (J) ChgB puncta per μm neurite.

K  Correlation between ChgB puncta per neuron and total neurite length for hippocampal (green) and striatal (red) neurons. Linear regression, inset shows goodness of fit ($r^2$) and significance (*P*-value) per group. Slopes are not significantly different, ANCOVA: $P = 0.97$.

L–P  Analysis of mouse cortex tissue expressing mCherry in axonal projections after stereotaxic viral injections in the thalamus and *post hoc* immunostained for ChgA as DCV marker. (L) Representative mouse brain slice with labeled axonal projections in cortical region (mCherry-filler, red) and immunostained for ChgA (green). Scale bar: 1 mm. (a, b) Zoom of axonal projections in (L), scale bars: 100 μm. (M–O) Representative confocal image and zoom of cortex tissue containing labeled axonal projections. Measurements of ChgA puncta (black) were performed within axonal projections (mask, yellow). Zoomed areas (N, O) are indicated in top, right image (M). Arrows (O) indicate ChgA staining measured as individual ChgA puncta. Scale bars: 10 μm (M), 5 μm (N, O). (P) ChgA puncta per μm axon from mouse tissue, compared to the distribution of ChgB puncta in axons of cultured hippocampal neurons (Fig 3K). Mann–Whitney *U*-test: ***$P = < 0.0008$. $N = 3$ mice (52 images of two cortical regions per mouse).

Data information: Bars show mean ± SEM, and gray dots represent measurements of individual single neurons. *N* numbers represent number of independent experiments, and the number of single neuron observations is in brackets.

**Figure 1.**

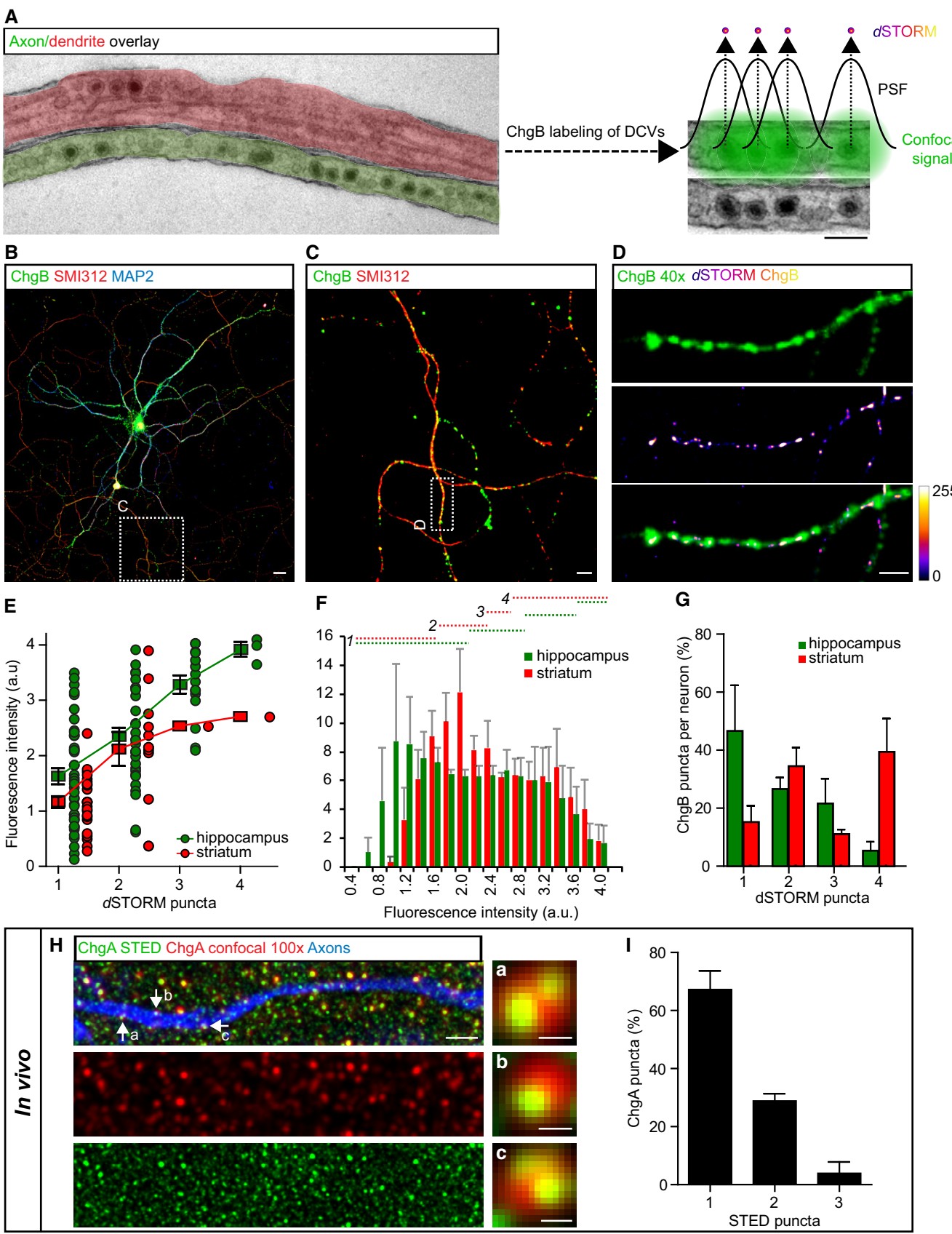

**Figure 2.**

single DCV and each DCV is loaded with a similar amount of ChgB and labeled with a similar number of antibodies. Imaging of Alexa 647-labeled ChgB using *d*STORM showed puncta of 60–200 nm in size (Fig 2D). A linear positive correlation between fluorescence intensity of ChgB and the number of *d*STORM puncta was observed in both hippocampal and striatal neurons (Fig 2E). The fluorescence intensity distribution of the total pool of ChgB puncta (Fig 2F) was divided in bins corresponding to the fluorescence representing the number of resolved *d*STORM puncta (Fig 2E and F, dotted lines). On average, per hippocampal neuron, 47% of the ChgB puncta had a fluorescence intensity representing one *d*STORM punctum, 27% an intensity representing two *d*STORM puncta, 22% representing three puncta, and 5.3% representing four puncta (Fig 2G, green). Per striatal neuron, on average, 15% of the ChgB puncta had a fluorescence intensity representing one *d*STORM punctum, 34% an intensity representing two *d*STORM puncta, 11% representing three puncta, and 39% representing four puncta (Fig 2G, red). From these findings, we conclude that the average 4,000 ChgB puncta in hippocampal neurons (Fig 1D) represent a total DCV pool of approximately 7,400 DCVs per neuron and the average 1,360 ChgB puncta in striatal neurons (Fig 1H) represent a total DCV pool of approximately 3,730 DCVs per neuron.

To resolve the number of DCVs in mouse neurons *in vivo*, we performed stimulated emission depletion (STED) microscopy on cortical brain slices with labeled thalamo-cortical axonal projections and stained for ChgA (Fig 1L–O). STED microscopy of ChgA resolved the confocal signal to single DCV localizations (Fig 2H). Within the axonal mask, on average, 67% of ChgA puncta overlaid with one STED punctum, 29% with two STED puncta, and 4% with three STED puncta (Fig 2I). We conclude that the average 0.53 ChgA puncta per μm axon (Fig 1P) represent approximately 0.72 DCVs per μm axon *in vivo*.

### DCVs are present in axonal, dendritic, and synaptic compartments and concentrate in dendrites

To study the subcellular location of DCVs, single mouse hippocampal neurons were cultured on glia islands and immunostained for

ChgB, β3-tubulin as neurite marker (axons and dendrites) and MAP2 as dendritic marker (Fig 3A). ChgB staining showed a punctate expression in MAP2-positive neurites (dendrites) and β3-tubulin-positive, MAP2-negative neurites (axons; Fig 3A–C). Co-staining of ChgB with VGLUT1 as a synapse marker showed DCVs in synapses but also at extra-synaptic sites (Fig 3D). ChgB staining also colocalized with β3-tubulin and showed approximately 50% overlap with MAP2 staining, indicating that ChgB staining originates from dendrites and axons (Fig 3F). VGLUT1 staining highly colocalized with ChgB (Fig 3G, Manders' coefficient 0.81), while ChgB staining overlapped for approximately 52% with VGLUT1 (Fig 3G, Manders' coefficient 0.52). This indicates that the majority of synapses contain DCVs. Colocalization of the staining for synaptic vesicle markers VGLUT1 and Synapsin was used as a positive control (Fig 3E and G) and VGLUT1 and Synapsin rotated 90° as negative control (Fig 3G). These data show that DCVs are present in axons and dendrites and in the majority of synapses of cultured hippocampal neurons.

To further study the distribution of DCVs in axonal and dendritic compartments, the number of ChgB puncta was calculated in all β3-tubulin-positive neurites and in MAP2-positive dendrites. On average, 4,000 ChgB puncta were present in all neurites per neuron, of which 50% was present in MAP2-positive dendrites (Fig 3H and I). When subtracting the ChgB puncta in MAP2-labeled dendrites from all β3-tubulin-positive neurites, approximately 50% of ChgB puncta were in β3-tubulin-positive, MAP2-negative axons (Fig 3H and I). The number of ChgB puncta per μm neurite was higher in dendrites than in axons (Fig 3K), because dendritic length per neuron was significantly shorter than axonal length (Fig 3J). The mean intensity per ChgB punctum was higher in dendritic areas (Fig 3L), and the intensity histogram showed a shift toward higher peak intensity for the dendritic population of ChgB puncta compared to the total pool (Fig 3M). To validate the distribution of ChgB puncta in axons and dendrites in single neurons, ChgB puncta were measured in SMI312-positive axons and MAP2-positive dendrites (Fig 3N and O). The number of ChgB puncta per μm neurite (Fig 3N) and the mean intensity per ChgB punctum were higher in dendrites compared to axons (Fig 3O). This indicates that dendrites, on average, contain

---

**Figure 2.  Quantification of total DCV pool *in vitro* and *in vivo* using super-resolution microscopy.**

A    Representation of resolution limit in confocal imaging and *d*STORM strategy to increase resolution. Electron micrograph of axon (green) and dendrite (red) containing DCVs, defined based on morphology. In conventional immunofluorescence confocal microscopy, individual DCVs cannot be separated (right zoom, green signal representing fluorescent signal using confocal). Super-resolution *d*STORM imaging resolves the point-spread function (PSF) resulting in single DCV localizations. Scale bars: 100 nm.

B, C    Representative confocal image and zoom of a single isolated hippocampal neuron (DIV14) immunostained for the endogenous DCV marker ChgB (green), axonal marker SMI312 (red), and dendritic marker MAP2 (blue). Areas of zoom (C) are indicated in (B). Scale bars: 20 μm (B), 5 μm (C).

D    ChgB immunoreactivity of zoomed area indicated in (C) (top panel). Reconstructed ChgB localizations of *d*STORM imaging (middle panel). Lower panel shows overlay of *d*STORM ChgB localizations and ChgB immunoreactivity (confocal, 40×). Calibration bar represents no (0) to high (255) clustering of *d*STORM localizations. Scale bar: 2 μm.

E    Fluorescence intensity of ChgB immunoreactivity puncta (confocal) with corresponding number of *d*STORM puncta for hippocampal (green) or striatal (red) cultured neurons. Squares show average ± SEM, and dots represent individual observations. Hippocampus: *N* = 2 cells (89 puncta), striatum: *N* = 3 cells (37 puncta).

F    Average histogram of fluorescence intensity of ChgB puncta (40× confocal) in neurons used for *d*STORM imaging. Dotted lines indicate cutoff values of fluorescence intensity corresponding 1, 2, 3, or 4 *d*STORM puncta. Data show average ± SEM. Hippocampus: *N* = 2 cells, striatum: *N* = 3 cells.

G    Percentage of ChgB puncta per neuron with corresponding fluorescence value of 1, 2, 3, or 4 *d*STORM puncta. Data show average ± SEM. Hippocampus: *N* = 2 cells, striatum: *N* = 3 cells.

H, I    Analysis of STED microscopy of mouse cortex tissue expressing mCherry in axonal projections after stereotaxic viral injections in the thalamus and *post hoc* immunostained for ChgA as DCV marker. (H) Representative confocal (100× magnification) image of mCherry-labeled axon (blue) with ChgA confocal signal (red) and corresponding super-resolution STED ChgA signal (green). Zoomed areas are indicated by arrows. Scale bar merged image (left): 1 μm, scale bars zooms: 100 nm. (I) Percentage of single ChgA puncta within mCherry-labeled axons with corresponding number of overlaying STED puncta. Data show average ± SEM. *N* = 2 mice (7 STED images, 70 single observations).

more intense ChgB puncta, suggesting that multiple DCVs cluster at specific locations in dendrites.

Axonal and dendritic markers showed high overlap in single cultured neurons (Fig EV1). Therefore, we assessed the distribution of ChgB puncta in a culture with sparse fluorescently labeled neurons (Fig EV2), which are not limited in space by a glial island. This offers better separation between axons and dendrites. The number of ChgB puncta per μm was similar to single neuron cultures (compare Fig 3K and P) and significantly higher in dendrites compared to axons, also similar to single neuron cultures (Fig 3P, mean dendrites: 0.87, mean axons: 0.31, *P* < 0.0001). The mean intensity per ChgB puncta was also higher in dendrites compared to axons (Fig 3Q). Together, these data show that DCVs are more enriched in dendrites compared to axons in cultured hippocampal neurons.

To test whether these estimates of DCV distribution in cultured neurons are comparable to mouse neurons *in vivo*, we labeled dentate gyrus (DG) granule neurons by stereotaxic injections of AAV expressing mCherry driven by the CAG promoter (Fig 3R). Hippocampal slices were prepared using the "magic cut" to visualize the mossy fibers (Bischofberger *et al*, 2006), and ChgA puncta were quantified in mCherry-positive axons and dendrites (Fig 3R). 0.42 ChgA puncta per μm dendrite and 0.46 ChgA puncta per μm axon were detected (Fig 3S). These data show that in DG granule neurons *in vivo,* DCVs are equally distributed between axons and dendrites.

### Average synapses contain two to three DCVs, at the periphery of the synaptic vesicle cluster

Dense-core vesicles are smaller than the resolution limit of confocal microscopy (Fig 2A, van de Bospoort *et al*, 2012). To study the synaptic localization of DCVs, we therefore used *d*STORM imaging of ChgB (Fig 4A–C) and electron microscopy (Fig 4D and E). *d*STORM imaging showed Alexa 647-labeled ChgB puncta at synapses, which localized to the border of VGLUT1-positive synaptic vesicle clusters (Fig 4C, zooms a–c). *d*STORM ChgB puncta were also found in axons and dendrites outside synapses (Fig 4C, middle panel, arrow).

Electron microscopy (EM) confirmed that DCVs are present at the border of synaptic vesicle clusters (Fig 4D and E). Fifty-four DCVs were found in 110 random presynaptic sections, approximately 0.45 DCV per section (Fig 4F, inset). Seventy percent of synaptic sections did not contain DCVs, 20% contained 1 DCV, and the remaining 10% contained multiple DCVs (Fig 4F). From the 110 random synaptic sections, only five showed a DCV in postsynaptic compartments. Using the volume of an average hippocampal bouton (Schikorski & Stevens, 1997) and the volume of synaptic EM sections, hippocampal synapses are predicted to contain on average 2.6 DCVs, with approximately 30% chance to find a DCV in a synaptic section (Fig 4F and see Materials and Methods). The average DCV diameter was 70 nm (Fig 4G), comparable with previous observations (Richardson, 1962; van de Bospoort *et al*, 2012). The distance to the active zone (AZ), measured from the center of the DCV to the center of the AZ, was on average 0.34 μm (Fig 4H and I), while the shortest distance from the center of the DCV to the AZ was on average 0.25 μm (Fig 4I; inset histogram). Compared to synaptic vesicles, DCVs were located further away from the AZ (Fig 4J), although we cannot exclude the presence of part of the active zone at a closer distance in another plane of the section. Approximately 80% of DCVs resided outside the synaptic vesicle cluster (Fig 4K and L), at close proximity to the border of the synaptic vesicle cluster (Fig 4M). Together, these results show that the

---

**Figure 3.  DCVs are present in axonal, dendritic, and synaptic compartments.**

A–C    Representative confocal images and zooms of a single isolated hippocampal neuron (DIV14) immunostained for the endogenous DCV marker ChgB (green), dendritic marker MAP2 (red), and β3-tubulin (blue) as neurite marker. Areas of zooms are indicated in (A). Closed arrows: dendritic ChgB staining, and open arrows: axonal ChgB staining. Scale bars: 20 μm (A), 5 μm (B, C).

D    Zoom of immunostaining for the excitatory synaptic vesicle protein VGLUT1 and DCV marker ChgB. Closed arrows: synaptic ChgB staining, and open arrows: extra-synaptic ChgB staining. Scale bar: 5 μm.

E    Zoom of immunostaining for two synaptic proteins VGLUT1 and Synapsin I/II. Scale bar: 5 μm.

F    Manders' coefficients of colocalization analysis between ChgB, β3-tubulin, and MAP2.

G    Manders' coefficients of colocalization analysis between ChgB and VGLUT1, with VGLUT1-Synapsin I/II colocalization as positive control and VGLUT1-Synapsin I/II rotated 90° as negative control.

H–M    Distribution of DCVs in single hippocampal neurons (DIV14) stained for ChgB, MAP2, and β3-tubulin. Dendrites were determined as MAP2-positive neurites, and axonal measurements were calculated by subtracting the dendritic population from the total pool measured in β3-tubulin-positive neurites. (H, I) Number of ChgB puncta per neuron in dendrites, axons, and the total pool (dendrites + axons). Wilcoxon matched-pairs signed rank test: n.s. = not significant, *P* = 0.75. (J) Total dendritic and axonal length (mm) per neuron. Wilcoxon matched-pairs signed rank test: ****P* = < 0.0001. (K) ChgB puncta per μm dendrite and axon per neuron. Lines next to individual data points show mean ± SEM. Wilcoxon matched-pairs signed rank test: ****P* = < 0.0001. (L) Mean ChgB fluorescence intensity per puncta in the total neuron (dendrites + axons) and in dendrites only. Wilcoxon matched-pairs signed rank test: ****P* = < 0.0001. (M) Histogram of fluorescence intensity of ChgB puncta in the total neuron (dendrites + axons) and in dendrites only.

N, O    Distribution of DCVs in single hippocampal neurons (DIV14) stained for ChgB, MAP2, and the axonal marker SMI312. ChgB puncta were measured in dendrites, determined as MAP2-positive neurites, and in axons, determined as SMI312-positive neurites. (N) ChgB puncta per μm dendrite and axon per neuron. Lines next to individual data points show mean ± SEM. Wilcoxon matched-pairs signed rank test: ****P* = < 0.0001. (O) Mean ChgB fluorescence intensity per puncta in dendrites and axons. Wilcoxon matched-pairs signed rank test: ****P* = < 0.0001.

P, Q    Distribution of DCVs in dendrites and axons of sparse-labeled network cultures. (P) ChgB puncta per μm dendrite or axon. Lines next to individual data points show mean ± SEM. Mann–Whitney *U*-test: ****P* = < 0.0001. (Q) Mean ChgB fluorescence intensity per puncta in dendrites and axons. Unpaired *t*-test: ****P* = < 0.0001.

R, S    Analysis of DCVs in axons and dendrites of mCherry-labeled mouse dentate gyrus (DG) granule cell tissue after stereotaxic viral injections and *post hoc* immunostained for ChgA as DCV marker. (R) Representative mouse brain slice with labeled DG granule cells (mCherry-filler, red) and immunostained for ChgA (green). ChgA puncta were measured in dendrites and axons at indicated areas. Scale bar: 250 μm. (S) ChgA puncta per μm dendrite or axon in DG granule cells. *N* = 1 mouse (six dendrite regions, seven axonal regions).

Data information: Bars show mean ± SEM, and dots represent measurements of individual single neurons or neuronal regions. *N* numbers represent number of experiments, and the number of (single) neuron observations is in brackets.

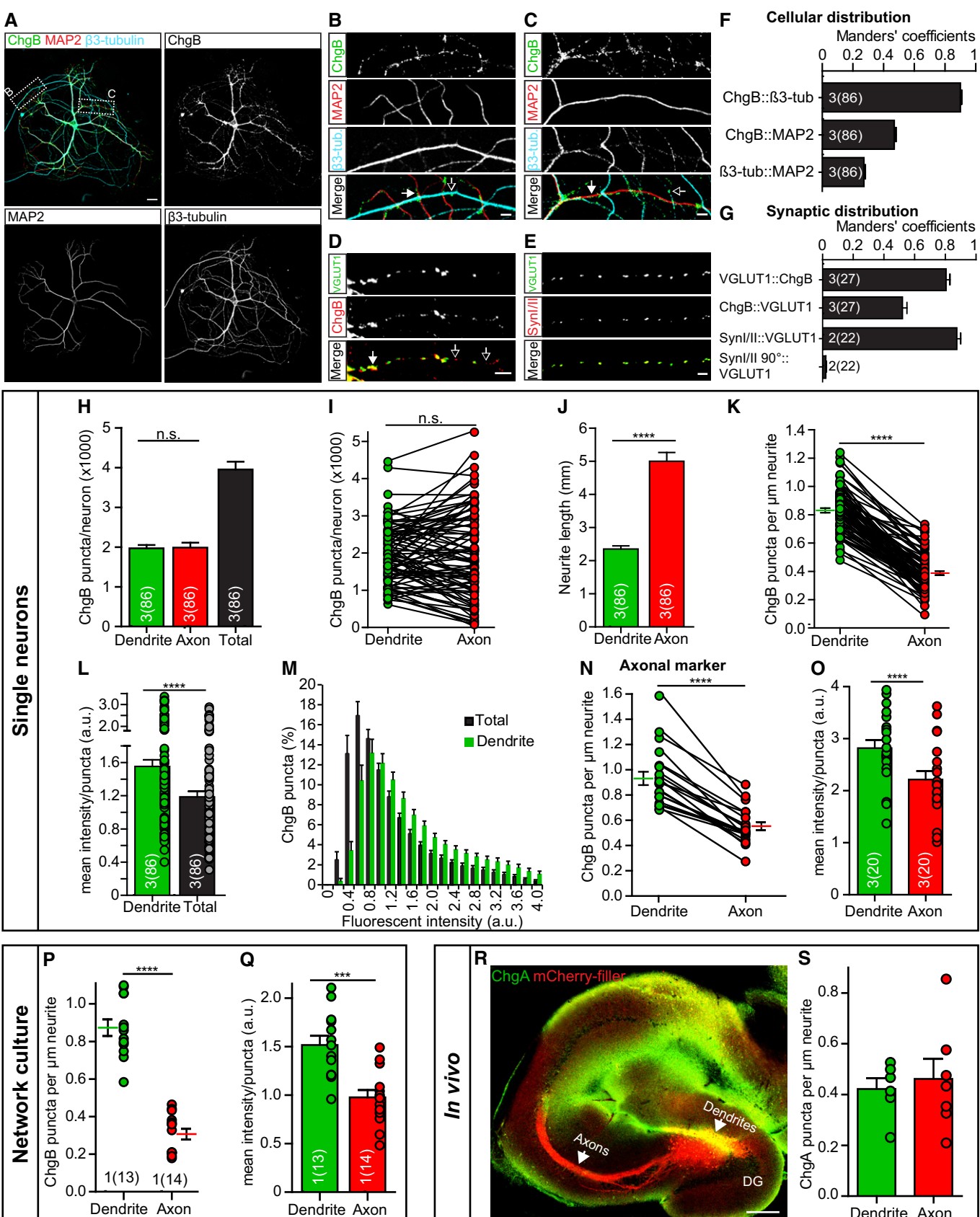

Figure 3.

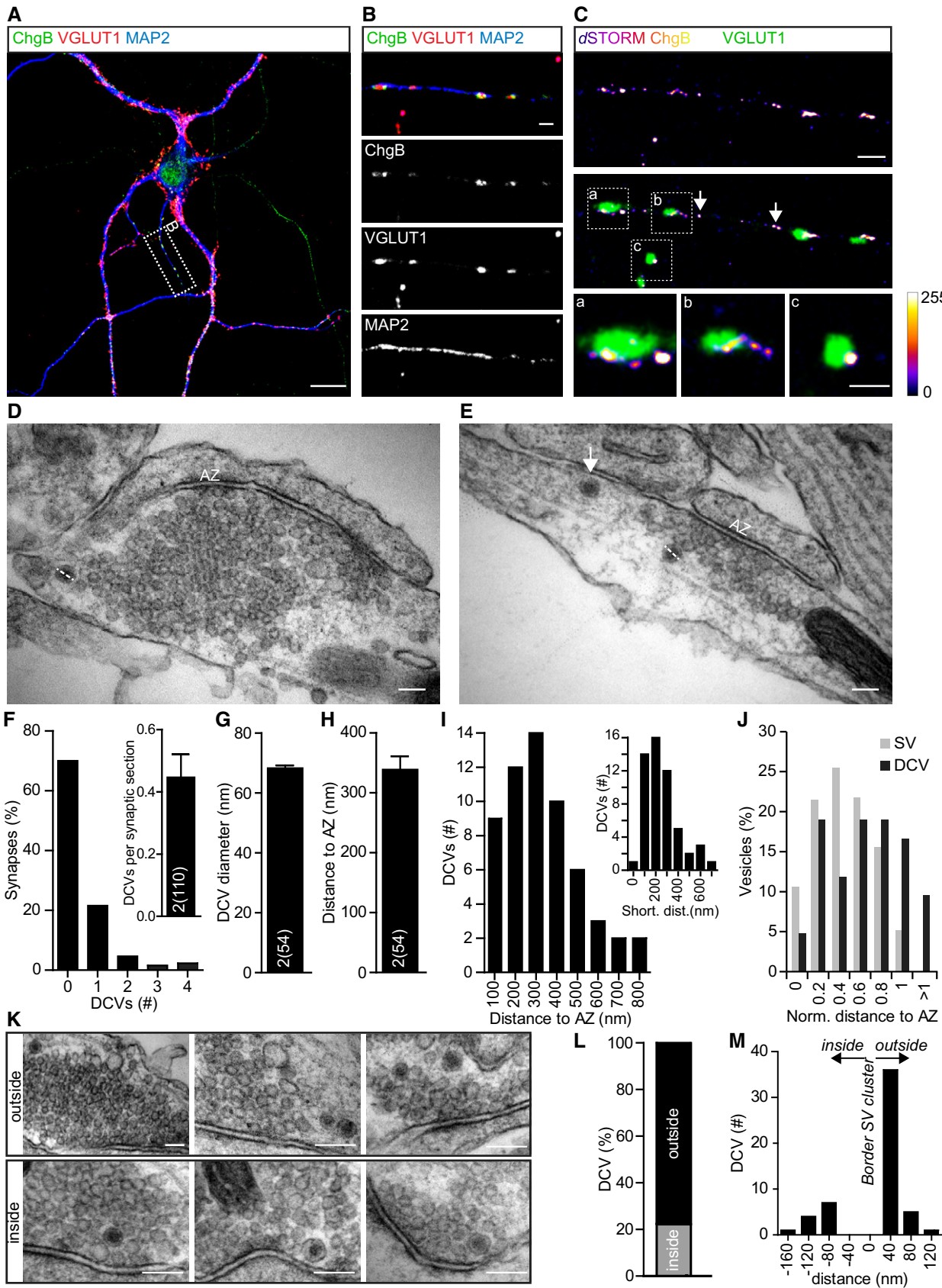

**Figure 4.**

**Figure 4. Synaptic DCVs localize to the border of the synaptic vesicle cluster.**

A, B   Representative confocal image and zoom of a single isolated hippocampal neuron (DIV14) immunostained for the endogenous DCV marker ChgB (green), synaptic protein VGLUT1 (red), and dendritic marker MAP2 (blue). Area of zoom (B) is indicated in (A). Scale bars: 20 μm (A), 2 μm (B).

C   Reconstructed ChgB localizations of *d*STORM imaging of zoomed area in (B) (top). Middle panel shows overlay between *d*STORM ChgB localizations and VGLUT1 immunofluorescence (confocal, 40×). Scale bar: 2 μm. Zooms of boxed areas containing synapses are visualized in lower panels (a–c). Arrows indicate extra-synaptic DCVs. Scale bar (a–c): 1 μm. Calibration bar represents no (0) to high (255) clustering of *d*STORM localizations.

D, E   Representative electron micrographs of synaptic sections containing DCVs. AZ indicates the active zone, and dashed lines represent DCV diameter measurements. Arrow indicates extra-synaptic DCV. Scale bars: 100 nm.

F   Histogram of number of DCVs in percentage of synapses. Inset: number of DCVs per synaptic section.

G   Average diameter of DCVs (nm).

H   Distance of DCVs to the active zone (nm), measured from center of DCV to the middle of the active zone.

I   Histogram of distance of DCVs to the middle of the active zone (nm). Inset: shortest distance of DCVs to the active zone (nm).

J   Histogram of normalized distance of percentage of synaptic vesicles (gray) to the active zone and corresponding distance of DCVs (black) to the active zone.

K   Example electron micrograph zooms of synapses containing DCVs outside (top) or inside (bottom) the synaptic vesicle cluster. Scale bars: 100 nm.

L   Percentage of DCVs outside or inside the synaptic vesicle cluster.

M   Histogram of distance (nm) of DCVs to the border of the synaptic vesicle cluster, negative values are inside, and positive values outside.

Data information: Bars show mean ± SEM. *N* numbers represent number of experiments, and the number of single observations is in brackets. *N* = 2 (110 synaptic sections with in total 54 DCVs).

average synapse/bouton contains two to three DCVs and that these DCVs are typically located at the periphery of the synaptic vesicle cluster (Figs 3D and 4).

### DCVs have a release probability of 1–6%, depending on the type of stimulation

To quantify DCV fusion events, we expressed the DCV reporter neuropeptide Y (NPY) fused to pH-sensitive EGFP (pHluorin; Palareti *et al*, 2016; Arora *et al*, 2017; van Keimpema *et al*, 2017; Emperador Melero *et al*, 2017; Farina *et al*, 2015). NPY-pHluorin is a reliable marker for neuropeptide release, irrespectively of which type of neuron or which neuropeptide is endogenously expressed in cultured mouse hippocampal neurons (Arora *et al*, 2017), mouse cortical neurons (van Keimpema *et al*, 2017), or human IPSC-derived neuronal micro-networks (Emperador Melero *et al*, 2017). This reporter was targeted to DCVs in all neurons and colocalized with the vast majority (90%) of ChgB-labeled DCVs (Fig 5A–C). The number of ChgB puncta and their mean intensity did not differ between infected and non-infected neurons (Fig 5D and E). NPY-pHluorin-infected neurons showed a strong correlation between number of ChgB puncta and total neurite length (Fig 5F), similar to non-infected neurons (see Fig 1K). These data show NPY-pHluorin expression did not affect the total number of DCVs per neuron.

Dense-core vesicle fusion is triggered by depolarization-induced $Ca^{2+}$-influx (Matsuda *et al*, 2009; de Wit *et al*, 2009; van de Bospoort *et al*, 2012; Hartmann *et al*, 2001; Balkowiec & Katz, 2002; Gärtner & Staiger, 2002). To assess which fraction of the total DCV pool can be released upon stimulation, DCV fusion was measured by expressing NPY-pHluorin in single hippocampal neurons. In the low pH environment of the DCV lumen, NPY-pHluorin is quenched (Fig 5A-a). A rapid increase in fluorescence is detected upon fusion pore opening when pHluorin de-quenches (Fig 5A-b), followed by a rapid decline in fluorescence once the cargo is fully released or upon fusion pore closure and reacidification of the vesicle lumen (Fig 5A-c). Different stimulation paradigms to trigger DCV fusion were compared: single action potentials (AP), a repetitive electrical stimulation train (16 bursts of 50 AP at 50 Hz) known to be efficient in triggering DCV fusion (van de Bospoort *et al*, 2012; Farina *et al*, 2015; Arora *et al*, 2017; van Keimpema *et al*, 2017; Emperador Melero *et al*, 2017; Hartmann *et al*, 2001), superfusion with 60 mM

KCl, or the calcium ionophore Ionomycin (5 μM; Fig 5G). Single AP stimulation resulted in a transient calcium influx (Fig 5G inset) but did not elicit DCV fusion (Fig 5H and I). Repetitive electrical stimulation (16 × 50 AP, 50 Hz) resulted in prolonged calcium influx and on average 120 DCV fusion events per cell (Fig 5G–I). Application of 60 mM KCl resulted in on average 80 DCV fusion events per cell during the stimulation (Fig 5G–I). Ionomycin application led to a slower calcium influx and delayed DCV fusion of approximately 30 DCVs per neuron (Fig 5G and H). These data show that DCV fusion occurs exclusively upon a repetitive or prolonged rise in intracellular calcium levels, either by entry through voltage-gated calcium channels during depolarization (electrical stimulation, 60 mM KCl) or by increasing the permeability of the membrane by application of a calcium ionophore.

The fraction of fusing DCVs over the total DCV pool (release probability) was calculated for the different types of stimulation. When computing the total DCV pool, a similar correlation was observed for NPY-pHluorin intensity and the number of ChgB-labeled DCVs resolved at the super-resolution level (Fig EV3), and was used to correct the total DCV pool (as in Fig 2). Upon repetitive electrical stimulation, a release probability of approximately 6% was measured (Fig 5J). Application of 60 mM KCl resulted in a release probability of approximately 4%, while Ionomycin elicited fusion of approximately 1% (Fig 5J).

To test possible differences in release capacity between excitatory and inhibitory neurons, viral infection of mKate2 driven by the CaMKII promoter was used as a marker for glutamatergic neurons (Benson *et al*, 1992; Wang *et al*, 2013). Upon repetitive electrical stimulation, both CaMKII-positive and CaMKII-negative neurons showed abundant DCV fusion with a release probability that was not significantly different between the two groups, although strong variation between neurons occurred (Fig 5K). Together, these data show that DCVs have a release probability of 1–6%, depending on the type of stimulation, which is similar in excitatory and inhibitory hippocampal neurons.

### DCVs fuse predominantly at axons

To test whether DCVs fuse preferentially at axons or dendrites, DCVs in network cultures were labeled with NPY-mCherry, which reports full cargo release events as complete disappearance of

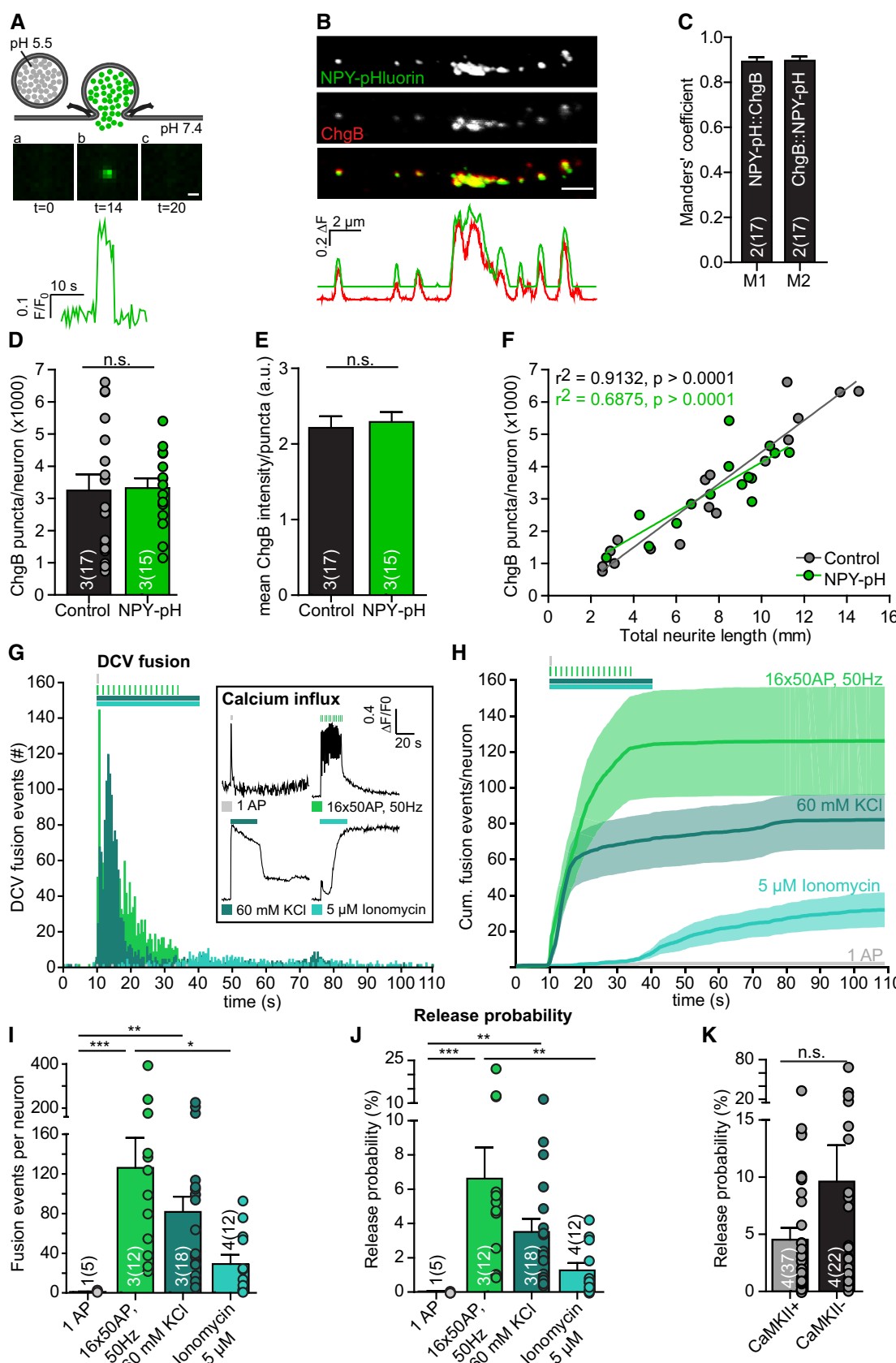

**Figure 5.**

**Figure 5.  DCV release probability upon different stimulations.**

A    Schematic representation of NPY-pHluorin as optical reporter for DCV fusion. Inset below shows $F/F_0$ of single DCV fusion event. Scale bar: 1 μm.
B    Zoomed confocal image of NPY-pHluorin expressing (green) hippocampal neuron (DIV14) immunostained for ChgB (red) and corresponding fluorescence intensity plot showing overlap between NPY-pHluorin and ChgB. Scale bar: 2 μm.
C    Manders' coefficients of colocalization between NPY-pHluorin and ChgB per neuron.
D    Total ChgB puncta per neuron in uninfected (control) and NPY-pHluorin-infected neurons (NPY-pH). Student's *t*-test: n.s. = not significant, $P = 0.90$.
E    Mean ChgB fluorescence intensity per puncta per neuron in uninfected (control) and NPY-pHluorin-infected neurons (NPY-pH). Student's *t*-test: n.s. = not significant, $P = 0.71$.
F    Correlation between ChgB puncta per neuron and total neurite length (mm) for uninfected control (gray) and NPY-pHluorin infected (green) neurons. Linear regression, inset shows goodness of fit ($r^2$) and significance (*P*-value) per group. Slopes are not significantly different, ANCOVA: $P = 0.19$.
G–J    Single isolated hippocampal neurons (DIV14) were labeled with NPY-pHluorin. DCV fusion events were quantified per neuron upon different stimulation paradigms. (G) Histogram of DCV fusion events per stimulation per time point. Stimulations include one action potential (AP) (gray), 16 bursts of 50 AP at 50 Hz (light green), superfusion with 60 mM KCl Tyrode's solution for 30 s. (dark green), and superfusion with 5 μM Ionomycin for 30 s. (blue). Inset: Typical $\Delta F/F_0$ trace of rise in intracellular calcium (Fluo5-AM) upon different stimulations. Traces are corrected for baseline (first 10 frames) and normalized. (H) Cumulative plot of fusion events per neuron for different stimulations. (I) DCV fusion events per neuron per stimulation. Kruskal–Wallis with Dunn's correction: *$P < 0.05$, **$P < 0.01$, ***$P < 0.001$. 1AP vs. 5 μM Ionomycin, 16 × 50 AP, 50 Hz vs. 60 mM KCl, and 60 mM KCl vs. 5 μM Ionomycin were non-significant, $P > 0.05$. (J) DCV release probability per neuron per stimulation. Kruskal–Wallis with Dunn's correction: **$P < 0.01$, ***$P < 0.001$. 1AP vs. 5 μM Ionomycin, 16 × 50 AP, 50 Hz vs. 60 mM KCl, and 60 mM KCl vs. 5 μM Ionomycin were non-significant, $P > 0.05$.
K    DCV release probability per neuron in CaMKII-positive and CaMKII-negative hippocampus neurons (DIV14) upon repetitive electrical stimulation (16 × 50 AP at 50 Hz). Mann–Whitney *U*-test: n.s. = not significant, $P = 0.12$.

Data information: Stimulation period is represented by colored bars above graphs. Bars show mean ± SEM. *N* numbers represent number of experiments, and the number of (single) neuron observations in brackets is represented as dots.

fluorescence signal upon fusion (Fig 6A), and neurons were sparse-labeled with membrane-bound mEGFP (Fig 6B). Axons were defined as small, spineless neurites positive for the axonal marker SMI312 in *post hoc* stainings, and dendrites as thick, spine-containing MAP2-positive neurites (Fig EV2). Upon maximal stimulation (16 bursts of 50 AP at 50 Hz, see above), DCV fusion events were observed in both axons and dendrites (Fig 6C and D). DCV fusion occurred predominantly at axons (Fig 6C and D). Axonal release was observed during the whole stimulation period, while dendritic release occurred predominantly in the first 10 s of stimulation with only a few events in the last bursts of stimulation (Fig 6C and D). Fusion onset was slightly earlier in axons compared to dendrites (Fig 6E; normalized cumulative plot). Per neuron, approximately 80% of the DCV fusion events occurred at axons (Fig 6F). Together, these data show that DCVs are fourfold more likely to fuse at axons.

# Discussion

This study characterized the number of DCVs present in individual mouse CNS neurons, their distribution, and release probability. Cultured hippocampal and striatal neurons contained a large pool of DCVs ranging from 1,400 to 18,000 DCVs per neuron, correlating with neurite length. DCVs were present in synapses, axons, and dendrites but were enriched in dendrites. Their distribution was slightly higher in axonal *in vivo* sections of mouse cortex compared to cultured neurons. DCVs showed a release probability of 1–6% depending on the stimulation, with repetitive electrical stimulation being most efficient, and fused predominantly at axons.

**Total DCV pool and DCV fusion quantified at single vesicle resolution indicate a release probability of 1–6%**

We used immunostainings and *d*STORM super-resolution imaging of ChgB and NPY-pHluorin on single cultured neurons to provide accurate estimations of the total DCV pool size in mammalian

neurons (Figs 2 and EV3). Using this method, a XY resolution of 20–50 nm was reached, but only 100 nm in the Z-direction and partial overlapping vesicles in the Z-dimension might not be fully reconstructed. Hence, although analysis of the intensity profiles confirmed that estimations of DCV numbers are generally accurate, some underestimation of the absolute number of DCVs cannot be excluded. Given these considerations, we conclude that cultured hippocampal neurons contain a total pool of 1,400–18,000 DCVs (Figs 1 and 2). Upon stimulation, 20–400 DCVs fuse, corresponding to 1–6% of the total pool (Fig 5). This secretion efficiency is similar to sporadic observations in axons and whole neurons (Shimojo *et al*, 2015; Xia *et al*, 2009; van de Bospoort *et al*, 2012; Farina *et al*, 2015; Emperador Melero *et al*, 2017).

The DCV release probability is comparable to secretory granule fusion in adrenal chromaffin cells and β-cells: Adrenal chromaffin cells that release catecholamines, adrenaline, and noradrenaline from secretory granules have an estimated total pool of approximately 2,000 vesicles with an RRP of 140, approximately 7% of the total pool (Sorensen, 2004; Voets *et al*, 1999) and murine pancreatic β-cells, which secrete insulin-containing granules, have a total pool of 10,000 insulin-containing vesicles, with an RRP of 200–300, 2–3% of the total pool (Dean, 1973; Olofsson *et al*, 2002). In cultured hippocampal synapses, a total pool of 100–200 synaptic vesicles per bouton has been estimated, with an RRP of approximately five vesicles and a recycling pool of 10–20 vesicles (Schikorski & Stevens, 1997; Neher, 2015; Rizzoli & Betz, 2005). The Calyx of Held contains approximately 500 release sites with each containing 250 synaptic vesicles, a recycling pool of 20,000–42,000 vesicles, and an RRP of 2,000–4,800 vesicles (De Lange *et al*, 2003; Yamashita *et al*, 2005; Fernández-Alfonso & Ryan, 2006). For synaptic vesicles, the release probability is defined as the reliability of an action potential to trigger synaptic vesicle fusion and is on average < 0.3, but ranging from < 0.05 to 0.9 for different nerve terminals (Zucker, 1973; Rosenmund *et al*, 1993; Murthy *et al*, 1997). When release probability is defined in the same way for DCVs, the release probability in the neurons studied here is zero (Fig 5), indicating a marked difference in the likeliness

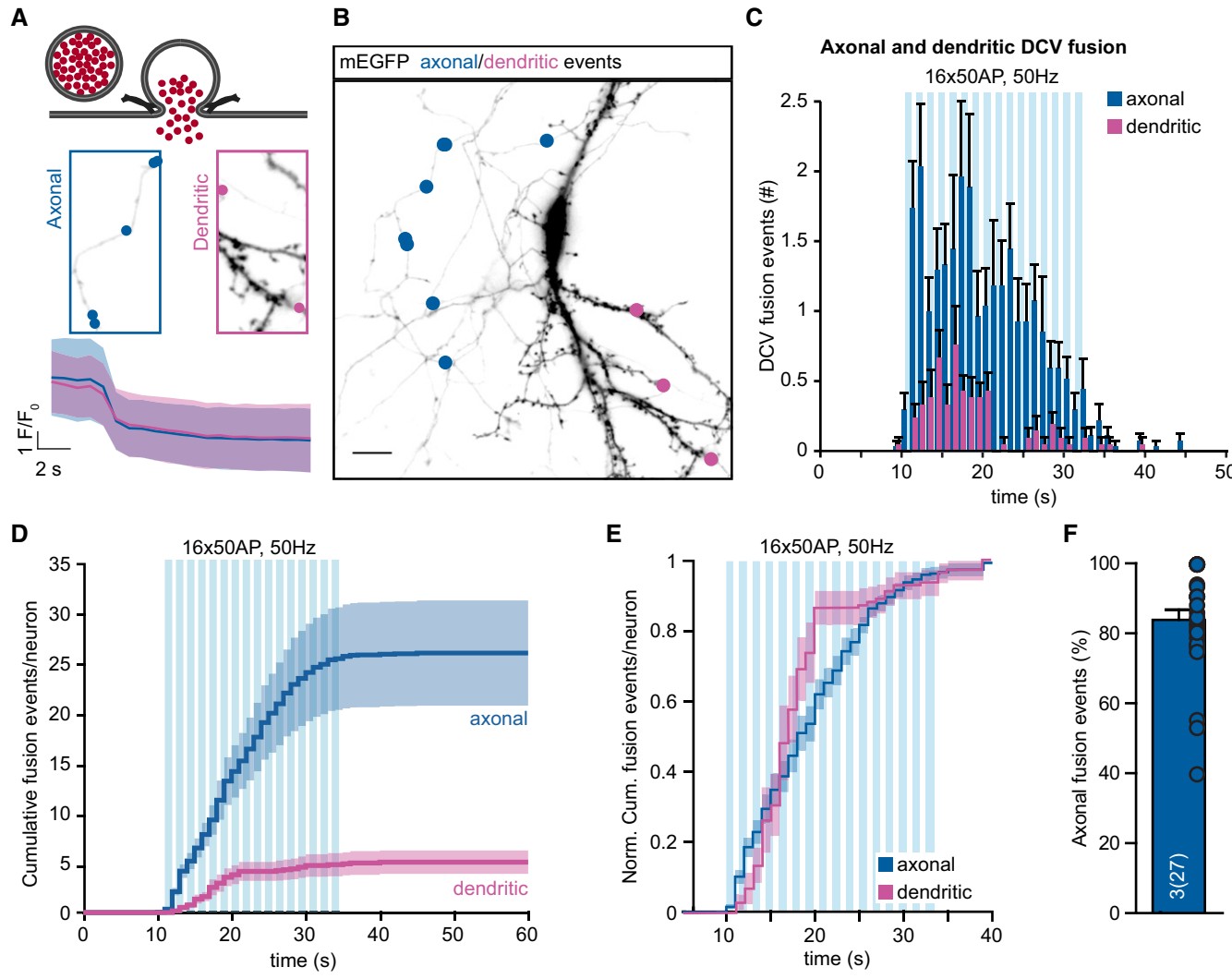

**Figure 6. DCV fusion occurs predominantly at axons.**

A  Schematic representation of NPY-mCherry as optical reporter for DCV fusion with full cargo release (top). Zoom of axonal and dendritic areas of (B), with location of fusion events (middle). Average fluorescence decrease upon NPY-mCherry-labeled DCV fusion in axons and dendrites (below).

B  Representative neuron labeled with membrane-targeted EGFP (mEGFP) (black) and localization of axonal (blue dots) or dendritic (purple dots) events upon stimulation. Scale bar: 20 μm.

C  Histogram of axonal (blue) and dendritic (purple) DCV fusion events upon repetitive electrical stimulation (16 × 50 AP at 50 Hz).

D  Cumulative plot of axonal or dendritic fusion events per neuron.

E  Normalized cumulative plot of axonal and dendritic fusion events per neuron.

F  Percentage of axonal fusion events per cell.

Data information: Trains of electrical stimulation are represented by blue bars. Bars show mean ± SEM. N numbers represent number of experiments, and the number of individual observations in brackets is represented as dots.

that the two secretory organelles will fuse and release their content upon mild stimulation. However, the release probability for neuropeptide secretion has been typically defined as the fraction of fusing DCVs during extensive stimulation over the total DCV pool, i.e., the equivalent of the "release pool" for synaptic vesicles (Shimojo *et al*, 2015; Xia *et al*, 2009; Emperador Melero *et al*, 2017). Similar repetitive stimulation depletes approximately 40% of the total synaptic vesicle pool [i.e., the release pool, (Fernández-Alfonso & Ryan, 2006; van Keimpema *et al*, 2017)]. Hence, also for stronger stimulation, synaptic vesicles are on average much more

likely to fuse than DCVs, probably indicating an evolutionary adaptation to faster, more efficient secretion.

For the different secretion pathways, different phases of release have been characterized. Chromaffin cells have a first initial burst, followed by a sustained phase (Voets *et al*, 1999) and glucose-stimulated insulin secretion consists of an early transient first phase and a sustained secondary phase (Olofsson *et al*, 2002). Neuronal DCV fusion upon repetitive electrical stimulation shows a peak in fusion only after several bursts of stimulation (Figs 5G and H, and 6C–E) and therefore does not seem to have a fast

component. Only upon many action potentials, more than 50 in the first burst, the first DCV starts to fuse. This observation confirms original findings in the submaxillary gland of the cat and the pig spleen *in vivo* (Andersson *et al*, 1982; Lundberg & Hökfelt, 1983; Lundberg *et al*, 1989), proposing "chemical frequency coding", i.e., the release of different chemical signals at different stimulation intensities.

## Several factors may explain why DCV fusion requires prolonged stimulation

The stimulation–secretion coupling for DCV fusion is remarkably different compared to other secretory pathways. The requirement for prolonged stimulation may be explained by (i) the translocation of DCVs to release sites, (ii) different calcium sensing principles for DCVs, and/or (iii) different calcium signals/sources required for DCV fusion.

(i) Unlike synaptic vesicles, DCVs are often highly mobile and generally not predocked at a release site. Hence, DCVs often need to translocate to a release site, which may require dissociation from the microtubule tracks that transport mobile DCVs (Zahn *et al*, 2004; Lo *et al*, 2011), and engage with the release machinery before exocytosis can occur. Synaptic DCVs are predominantly located at the border of the synaptic vesicle cluster (Fig 4), and although DCVs do not accumulate at synapses, approximately 60% of fusion events occur at the synapse (van de Bospoort *et al*, 2012; Farina *et al*, 2015). This preferential synaptic fusion is much slower than synaptic vesicle fusion, consistent with the observation that synaptic DCVs are not docked and therefore not in close contact with the release machinery. The synaptic vesicle cluster may act as a barrier for DCVs to reach their release site.

(ii) Tight spatial coupling between synaptic vesicles and voltage-gated calcium channels regulates fast exocytosis (Zucker & Fogelson, 1986; Llinás *et al*, 1992). High levels of prolonged calcium influx are probably required before calcium reaches DCVs and can activate DCV calcium sensors (Fig 5G). Although it is well established that DCV fusion critically depends on $Ca^{2+}$ (Hartmann *et al*, 2001; Balkowiec & Katz, 2002; de Wit *et al*, 2009), it remains unclear how $Ca^{2+}$ actually activates fusion of vesicles that are not yet docked and are therefore not yet engaged with the components of the fusion machinery known to reside at the plasma membrane.

(iii) Presynaptic functions are controlled by the calcium content of the axonal endoplasmic reticulum (ER) which regulates changes in synaptic cytosolic calcium levels during prolonged activity (de Juan-Sanz *et al*, 2017). In *Drosophila* motor neurons, it was shown that ryanodine receptor (RyR)-mediated calcium release from presynaptic ER and resulting calmodulin kinase II activation are required for post-tetanic potentiation of neuropeptide secretion (Shakiryanova *et al*, 2007). Furthermore, the activation of cAMP-dependent protein kinase may be required for DCV fusion (Shakiryanova *et al*, 2011). Therefore, calcium-induced calcium influx from internal stores, which requires prolonged activity, may be a unique aspect of DCV fusion, which discriminates this type of chemical signaling from synaptic transmission (Nizami *et al*, 2010; Shakiryanova *et al*, 2007, 2011).

## Mammalian DCVs do not accumulate at synapses

In *Drosophila* motor neurons, bidirectional transported DCVs are captured at nerve terminals upon activity, providing a constant availability of synaptic DCVs (Wong *et al*, 2012). In this study, electron microscopy and *d*STORM imaging showed that DCVs are localized at the perimeter of the synaptic vesicle cluster in resting mammalian CNS neurons, but do not accumulate at synapses (Fig 4). In chronic experiments (chronic disinhibition) and upon 90-s depolarization with high $K^+$, synaptic capture has also been described for neurotrypsin containing DCVs in mammalian neurons (Frischknecht *et al*, 2008). Hence, synaptic DCV capture may also exist in mammalian neurons, but occurs mostly during/after stimulation and requires strong/permanent stimulation, and in resting neurons, DCVs are apparently not accumulated.

## DCVs are equally distributed over axons and dendrites but fuse predominantly in axons

Dense-core vesicles are present in axons and dendrites in cultured hippocampal neurons (Fig 3). This is in line with previous studies, which have shown that KIF1 motors transport DCVs to axons and dendrites (Lipka *et al*, 2016; Zahn *et al*, 2004). The current study suggests that a similar number of DCVs ends up in axons and dendrites (Fig 3). The intensity of ChgB puncta was on average higher in dendrites, suggesting that multiple DCVs may cluster at certain locations in dendrites and the total population of dendritic DCVs may be somewhat underestimated. Furthermore, super-resolution microscopy showed a linear relation between intensity of ChgB puncta and the number of DCVs (Fig 2), indicating that most likely dendrites contain more DCVs than axons. Conversely, in cultured neurons, axons tend to run along and wrap around dendrites (Fig EV1) and some ChgB signal may therefore be incorrectly labeled as dendritic. Analysis of sparse-labeled network cultures provided better separation of axons from dendrites and led to the same conclusion as in single cultured neurons (Fig 3) that DCVs have a higher density in dendrites compared to axons.

Interestingly, the majority of DCV fusion events occurred at axons in the neurons studied here (Fig 6). Hence, axonal DCVs have a much higher release probability compared to dendritic DCVs. This is markedly different from a few specialized neurons in the mammalian brain, such as paraventricular and supraoptic nuclei (PVN/SON; Ludwig & Leng, 2006) or raphe nucleus neurons. However, for the majority of neurons, this might be explained by the availability of SNARE proteins, and other proteins involved in membrane fusion in axons. Furthermore, it has previously been shown that BDNF-containing DCVs show different fusion properties at axons, where incomplete cargo release was observed, while full fusion events occurred at dendrites (Dean *et al*, 2009; Matsuda *et al*, 2009). However, the NPY-mCherry reporter we used in this study to distinguish between axonal and dendritic release (Fig 6) did not show differences in fusion characteristics between axons and dendrites. NPY-mCherry detects full cargo release and does not discriminate between different types of release. In conclusion, in the neurons studied here, DCV fusion events occur predominantly at the axon, while DCVs accumulate more in dendrites, indicating strong differences in the probability of release based on the cellular localization of a DCV.

# Materials and Methods

## Animals

Embryonic day (E) 18.5 C57BL/6 mouse embryos were used for neuronal cultures. Animals were housed and bred according to institutional and Dutch governmental guidelines (DEC-FGA 11-03 and AVD112002017824).

## Primary neuronal cultures

Dissociated hippocampal or striatal neuron cultures were prepared from E18.5 C57BL/6 mouse embryos. Cerebral cortices were dissected free of meninges in Hanks' balanced salt solution (Sigma, H9394) supplemented with 10 mM HEPES (Gibco, 15630-056). The hippocampi or striata were isolated from the tissue and digested with 0.25% trypsin (Gibco, 15090-046) in Hanks' HEPES for 20 min at 37°C. Hippocampi were washed three times with Hanks' HEPES and triturated with fire-polished glass pipettes. Dissociated neurons were counted and plated in neurobasal medium (Gibco, 21103-049) supplemented with 2% B-27 (Gibco, 17504-044), 1.8% HEPES, 0.25% Glutamax (Gibco, 35050-038), and 0.1% penicillin–streptomycin (Gibco, 15140-122). To obtain single neuron micro-island cultures, hippocampal neurons were plated in 12-well plates at a density of 1,400–1,500 cells/well on 18-mm glass coverslips containing micro-islands of rat glia. For glia preparations, newborn pups from female Wistar rats were used. Micro-islands were generated as described previously (Meijer *et al*, 2015) by plating 8,000/well rat glia on UV-sterilized agarose (Type II-A; Sigma, A9918)-coated etched glass coverslips stamped with a mixture of 0.1 mg/ml poly-D-lysine (Sigma, P6407), 0.7 mg/ml rat tail collagen (BD Biosciences, 354236), and 10 mM acetic acid (Sigma, 45731).

For super-resolution microscopy experiments, coverslips were coated with 0.01% poly-L-ornithine (Sigma, P4957) and 2.5 μg/ml laminin (Sigma, L2020) diluted in Dulbecco's phosphate-buffered saline (DPBS; Gibco, 14190-250) overnight at room temperature (RT). Dissociated hippocampal neurons or striatum neurons were plated in low density (1,000–3,000 neurons) on coated coverslip and placed in close proximity to astrocytes previously grown on the bottom of 12-well plates to provide glia support (Kaech & Banker, 2006).

## Constructs

NPY-pHluorin or NPY-mCherry were generated by replacing Venus in NPY-Venus (Nagai *et al*, 2002) with super-ecliptic pHluorin or mCherry. CaMKII-mKate2 was a kind gift of Oleksandr Shcheglovitov (University of Utah School of Medicine, Salt Lake City, USA). Synapsin-mCherry was a kind gift of Dr. A. Jeromin (Allen Brain Institute, Seattle, USA) and Synapsin-ECFP was obtained by replacing mCherry with ECFP.

A tetracycline-controlled Tet-On gene expression system containing NPY-mCherry and membrane-bound mEGFP was used to determine the axonal or dendritic localization of DCV fusion. Membrane-bound mEGFP was generated by the addition of the last 20 amino acids (KMSKDGKKKKKKSKTKCVIM) of K-Ras4B, representing a canonical palmitoylation signal (Welman *et al*, 2000).

All constructs were sequence-verified and subcloned into pLenti vectors, and viral particles were produced (Naldini *et al*, 1996).

Adeno-associated virus particles of CAG-mCherry were used for stereotaxic injections of mice.

## Infection

For DCV fusion experiments neuronal cultures were infected with lentiviral particles encoding for NPY-pHluorin at DIV 9–10. CaMKII-mKate2 was added at DIV7.

To determine axonal or dendritic localization of release, WT hippocampal neurons were infected in suspension with TRE(pr)hNPYmCherry-IRES2-KrasGFP-pSyn(pr)rtTA2 and incubated for 2 h at 37°C, 5% CO$_2$. Cells were centrifuged for 5 min at 800 rpm, washed three times in DMEM, resuspended in neurobasal medium supplemented with 2% B-27, and plated (5,000 neurons/well) on coverslips containing 25,000 WT hippocampal neurons per well. 2 μg/ml doxycycline hyclate (Sigma, D9891) was added at DIV 10–11, and cells were imaged at DIV 16–17.

## Immunohistochemistry of mouse brain tissue

### Stereotaxic injection

For viral injection (Wimmer *et al*, 2004), mice were injected s.c. with 0.1 mg/kg buprenorphine, anaesthetized with a constant flow of 1–2% isofluorane, and placed in a stereotaxic head holder (Kopf). Before cutting of the skin, xylocain solution (2%) was injected subcutaneously. A small hole in the skull was made with a dental drill over the injection site. Then, for thalamic injections, 1 μl AAV particles was slowly injected into the right medio-dorsal thalamus (Coordinates from bregma: $x = 1.13$, $y = -0.82$, $z = -3.28$). For stainings in the dentate gyrus, animals were injected with 100 nl AAV particles (coordinates: AP: −3.16 ML + −2.2, DV: 2.3). After surgery, mice received carprofen (Rimadyl, 5 mg/kg; Pfizer). Mice were single-housed after surgery.

### Immunohistochemistry

For immunohistochemistry, mice were transcardially perfused with 4% paraformaldehyde (PFA) in phosphate-buffered saline (PBS). The brains were postfixed overnight in 4% PFA and then sliced into 70-μm-thick sections and stored in PBS. To preserve dentate gyrus granule cells with dendrites and mossy fibers in the same slice, brains were sliced horizontally under an angel as described elsewhere (Bischofberger *et al*, 2006). For increased permeabilization, the slices were treated with 1% sodium dodecyl sulfate (SDS) in PBS for 5 min before antibody staining as described previously (Knabbe *et al*, 2018). For antibody staining, slices were incubated for 1 h in PBS containing 5% normal goat serum (NGS), 1% bovine serum albumin (BSA), 1% cold fish gelatin, and 0.5% Triton X-100. Incubations with primary antibody were done overnight at 4°C. The secondary antibody was incubated for 1–2 h at room temperature. Primary antibody was chromogranin A (SySy, 259003; 1:500). Alexa Fluor antibody from Invitrogen was used as secondary antibody in a dilution of 1:500. The slices were mounted in SlowFade Gold (Life Technologies) and imaged on a Leica SP8 inverted confocal microscope with a 63× oil immersion objective (NA = 1.4) and maximal resolution in *x*, *y* and *z* (0.08 × 0.08 × 0.3 μm).

## Immunocytochemistry cultured neurons

Neuronal cultures were fixed in 3.7% formaldehyde (Electron Microscopies Sciences, 15680) in PBS, pH 7.4, for 20 min at RT. After several washing steps with PBS, cells were permeabilized for 5 min with 0.5% Triton X-100 (Fisher Chemical, T/3751/08)–PBS and subsequently incubated for 30 min with PBS containing 2% normal goat serum (Gibco, 16210-072) and 0.1% Triton X-100 to block nonspecific binding. Incubations with primary and secondary antibodies were performed for 1 h at RT with PBS washing steps in between.

### Primary antibodies used were as follows

Polyclonal rabbit chromogranin B (1:500; SySy 25103), monoclonal mouse β3-tubulin (1:500; Millipore MAB1637), polyclonal chicken MAP2 (1:10,000; Abcam ab5392), monoclonal mouse GFAP (1:1,000; Sigma G3893), monoclonal mouse SMI-312 (1:5,000; Covance), polyclonal guinea pig VGLUT1 (1:1,000; Millipore AB5905), polyclonal rabbit VGAT (1:500; SySy 131002), and polyclonal rabbit Synapsin I&II (1:1,000; E028).

Alexa Fluor conjugated secondary antibodies were from Invitrogen (1:1,000). Coverslips were washed again and mounted with Mowiol 4-88 (Sigma, 81381) and examined on a confocal microscope [Nikon A1R with LU4A laser unit or Zeiss LSM 510 confocal microscope using a 40× oil immersion objective (NA = 1.3)]. Z-stacks (5 steps of 0.5 μm) were acquired, and resulting maximum projection images were used for analysis. A 60× oil immersion objective (NA = 1.4) was used for detailed zooms. To count the number of VGLUT1$^+$ or VGAT1$^+$ neurons in hippocampal or striatal cultures, coverslips were scanned to ensure every neuron was captured. Confocal settings were kept constant for all scans within an experiment.

## Electron microscopy

Hippocampal autaptic cultured neurons were fixed for 60 min at room temperature with 2.5% glutaraldehyde in 0.1 M cacodylate buffer (pH 7.4), postfixed for 1 h at room temperature with 1% $OsO_4$/1% $K_4Ru(CN)_6$ in double distilled water. Following dehydration through a series of increasing ethanol concentrations, cells were embedded in Epon and polymerized for 24 h at 65°C. After polymerization of the Epon, the coverslip was removed by alternately dipping in liquid nitrogen and hot water. Cells were selected by observing the Epon-embedded culture under the light microscope, and mounted on prepolymerized Epon blocks for thin sectioning. Ultrathin sections (approximately 70 nM) were cut parallel to the cell monolayer and collected on single-slot, formvar-coated copper grids, and stained in uranyl acetate and lead citrate.

Synapses were selected at low magnification using a JEOL 1010 electron microscope. All analyses were performed on single ultrathin sections of randomly selected synapses. Digital images of synapses were taken at 100,000× magnification using iTEM software (EMSIS, Germany). For all morphological analyses, we selected only synapses with intact synaptic plasma membranes with a recognizable pre- and postsynaptic density and clear synaptic vesicle membranes. DCVs were recognized as an electron dense core surrounded by a vesicular membrane. Measurements were performed using ImageJ software.

To calculate the average number of DCVs per synapse, the volume of a typical synapse (assumed to be a sphere) was divided by volume of the synaptic area of a typical slice (assumed to be a cylinder) and was calculated as follows:

$$\text{Volume cultured hippocampal synapse} = \frac{4}{3}\pi r^3 = 0.122\,\mu m^3$$

(Schikorski & Stevens, 1997), $r = 0.307$

$$\text{Volume synaptic slice} = r^2\pi \times d = 0.02073\,\mu m^3,$$

$d$ = thickness slice = 70 nm

$$\text{DCVs per synapse} = \frac{\text{Volume synapse}}{\text{Volume synaptic slice}}$$
$$\times \text{ average DCV per synapse}$$
$$= 5.8851 \times 0.45 = 2.633$$

Chance of finding DCV in synaptic section :
$$\frac{\text{Thickness synaptic slice}}{\text{Diameter/thickness synapse}}$$
$$\times \text{DCVs per synapse} = 0.304$$

## Direct stochastic optical reconstruction microscopy (*d*STORM) acquisition

PFA fixation and staining protocol for *d*STORM imaging were optimized according to (Whelan & Bell, 2015) and included 12 min fixation with 3.7% PFA-PBS (37°C followed by several PBS washes lasting 30 s, 1, 5, 10, and 15 min). Normal immunocytochemistry protocol was followed using 5% BSA-PBS as blocking buffer. Coverslips were stored in PBS until image acquisition.

Coverslips were incubated with fluorescent microspheres (Invitrogen constellation microspheres c14837) and placed in a closed imaging chamber containing photo-switchable buffers with oxygen scavengers consisting of 50 mM MEA-HCl (Sigma M6500), enzyme solution [40 μg/ml catalase (Sigma C1345), 4 mM TCEP (Sigma C4706), 25 mM KCl, 0.5 mg/ml glucose oxidase (Sigma G2133), 20 mM Tris–HCl pH 7.5, 50% glycerin)], and a glucose–glycerin base solution (10% glycerin, 10% glucose).

Imaging was performed on a Nikon Ti-E Eclipse inverted microscope system with LU4A Laser unit and an EMCCD Camera (Andor DU-897) equipped with 647-nm laser, a perfect focus system and an astigmatic lens for 3D localizations (Nikon). A 100× oil objective lens (NA 1.49) and appropriate filtersets were used for *d*STORM acquisition. NIS elements software (version 4.30) was used to control acquisition in streaming mode of 256 × 256 pixel region of interest. 8,000 frames per region were acquired. After transferring the fluorophores to the OFF state, laser power was adjusted to keep the number of the stochastically activated molecules constant and well separated during the acquisition.

## Stimulated emission depletion microscopy (STED) acquisition

Stimulated emission depletion microscopy (STED) was performed on a Leica TCS SP8 STED 3× microscope, Leica Microsystems (Wetzlar, Germany). Samples were irradiated with a pulsed white light laser at wavelengths 499 and 590 nm. A continuous wave STED laser line at a wavelength of 592 nm, Leica Microsystems, was used for depletion of the 488 nm fluorophore, reaching a lateral resolution of ~70 nm. The signal was detected using a gated hybrid

detector (HyD), Leica Microsystems, in photon-counting mode. STED images were acquired using a dedicated oil objective with 100× magnification and a numerical aperture of 1.4 (Leica Microsystems). A Z-stack was made with a step size of 150 nm and pixel size of 19 × 19 nm, optimized using Nyquist Calculator from SVI (Scientific Volume Imaging, Hilversum, the Netherlands). Finally, deconvolution was performed with Huygens Professional (SVI).

## Live imaging

Live imaging experiments were performed on a Nikon Ti-E Eclipse inverted microscope system fitted with a Confocal A1R (LU4A Laser) unit and an EMCCD (Andor DU-897). The inverted microscope together with the EMCCD was used for live imaging using the LU4A laser unit with a 40× oil objective (NA 1.3) and appropriate filtersets. NIS elements software (version 4.30) controlled the microscope and image acquisition.

Coverslips were placed in an imaging chamber and perfused with Tyrode's solution (119 mM NaCl, 2.5 mM KCl, 2 mM $CaCl_2 \cdot 2H_2O$, 2 mM $MgCl_2 \cdot 6H_2O$, 25 mM HEPES, and 30 mM glucose·$H_2O$, pH 7.4, mOsmol 280). Isolated single neurons on glial islands were selected for acquisition. Time-lapse (2 Hz) recordings consisted of 30 s baseline recordings followed by stimulation. Electrical field stimulation was applied through parallel platinum electrodes powered by a stimulus isolator (WPI A385) delivering 30-mA, 1-ms pulses, regulated by a Master-8 pulse generator (A.M.P.I.) providing 1 action potential (AP) or 16 trains of 50 APs at 50 Hz with a 0.5-s interval. Chemical stimulations were applied through glass capillaries placed in close proximity the cell by gravity flow and included 60 mM KCL Tyrode's solution (61.5 mM NaCl, 60 mM KCl, 2 mM $CaCl_2 \cdot 2H_2O$, 2 mM $MgCl_2 \cdot 6H_2O$, 25 mM HEPES and 30 mM glucose·$H_2O$, pH 7.4, mOsmol 280) or 5 μm Ionomycin (Fisher BioReagent) dissolved in normal Tyrode's solution. Intracellular pH was neutralized by barrel application of normal Tyrode's solution containing 50 mM $NH_4Cl$, which replaced NaCl on an equimolar basis in the solution. To define calcium influx profiles upon stimulation, neurons were incubated for 15 min with 1 μM Fluo-5F-AM (Molecular Probes, F14222; stock in DMSO). All experiments were performed at RT (20–24°C), and no NMDA receptor or AMPA receptor antagonists were added.

## Imaging analysis

### DCV total pool size
Neuronal morphology and DCV numbers were analyzed using automated image analysis software SynD (Schmitz *et al*, 2011). Synapse detection settings were optimized to measure ChgA, ChgB puncta, or NPY-pHluorin signal and kept constant for the corresponding dataset.

### Colocalization
For colocalization analysis of different markers, morphological masks were drawn using SynD (Schmitz *et al*, 2011) and imported in ImageJ to remove background fluorescence. Colocalization was measured in ImageJ with JACoP (Bolte & Cordelieres, 2006). Thresholds were set manually to correct for background.

### DCV fusion
Dense-core vesicle fusion events were analyzed in stacks of time-lapse recordings (2 Hz, 512 × 512 pixels). In ImageJ DCV, fusion events were manually selected and fluorescence traces were measured in a circular 4 × 4 pixel ROI (1.56 × 1.56 μm). NPY-pHluorin events were defined by a sudden increase in fluorescence, and NPY-mCherry events were defined as a sudden decrease in fluorescence. Resulting fluorescence traces were loaded in a custom-built Matlab program where the traces were expressed as fluorescence change ($\Delta F$) compared to initial fluorescence ($F_0$) obtained by averaging the first 10 frames of the time-lapse recording. Fusion events were automatically detected and included when fluorescence showed a sudden increase (NPY-pHluorin) or a sudden decrease (NPY-mCherry) two standard deviations above or below $F_0$. Start of a fusion event was defined as the first frame above 2 × SD of $F_0$ and end of the fusion event as the first frame below this threshold.

### Calcium imaging
Calcium measurements were performed in ImageJ. Five neurite-located ROIs (4 × 4 pixels) and a background ROI were measured per neuron. Normalized $\Delta F/F_0$ data were calculated per cell after background subtraction.

### dSTORM
dSTORM images were reconstructed using the open-source ImageJ plugin ThunderSTORM (Ovesný *et al*, 2014). 3D calibration was performed on recordings of fluorescence microspheres, and single molecule localizations were reconstructed using Wavelet B-Spline filtering and 3D astigmatism. Postprocessing was performed to remove localizations with unrealistic parameters, and drift correction was performed using fiducial markers.

To estimate the number of DCVs per fluorescence puncta measured with confocal microscopy, an overlay was reconstructed between dSTORM regions and corresponding confocal images (40×). Intensity of ChgB puncta was measured in confocal images, and the number of reconstructed ChgB clusters with dSTORM was counted in the corresponding area. SynD analysis was performed on confocal recordings (40×) of the corresponding neurons used for dSTORM reconstruction to evaluate, based on intensity of ChgB puncta, the underrepresentation of pool size measurements using SynD. The fluorescence intensity distribution of the total pool of ChgB puncta was divided based on the number of resolved dSTORM puncta. Cutoff measurements were set as the intermediate between the average fluorescence intensity of 1–2, 2–3, and 3–4 dSTORM puncta.

### STED
To estimate the number of DCVs per fluorescence punctum measured with confocal microscopy, an overlay was reconstructed between the deconvoluted STED image of ChgA and corresponding deconvoluted confocal image (100× magnification). The number of STED puncta within a single ChgA confocal puncta was counted.

The soma was always excluded from analysis.

## Statistics

Shapiro–Wilk test was used to assess distribution normality. When assumptions of normality or homogeneity of variances were met, parametric tests were used: Student's *t*-test or one-way ANOVA (Tukey as *post hoc* test). Otherwise, non-parametric tests used were Mann–Whitney *U*-test for two independent groups, or Kruskal–Wallis with Dunn's correction for multiple groups. Wilcoxon

matched-pairs signed rank test was used for paired data, and slopes of linear regressions were tested using ANCOVA. Data are plotted as mean with standard error of the mean; *N* represents number of independent experiments, and *n* represents the number of cells.

**Expanded View** for this article is available online.

## Acknowledgements

The authors thank Prof. Thomas Kuner and Dr. Johannes Knabbe (Heidelberg) for *in vivo* immunostaining data; Robbert Zalm for cloning and producing viral particles, Frank den Oudsten and Desiree Schut for producing glia feeders and primary culture assistance, Joke Wortel for animal breeding, Joost Hoetjes for genotyping, Rien Dekker for expert assistance in electron microscopy, Jeroen Kole (VUmc imaging facility) for help with STED imaging, and members of the CNCR DCV project team for fruitful discussions. EM analysis was performed at the VU/VUmc EM facility (ZonMW 91111009). This work is supported by an ERC Advanced Grant (322966) of the European Union (to M.V.), the Agentschap NL (NeuroBasic PharmaPhenomics, FES0908 to M.V.), and Education, Audiovisual and Culture Executive Agency (EACEA; Erasmus Mundus Joint Doctorate grant EU 2011-1632/001-001-EMJD to M.F.).

## Author contributions

CMP, RFT, and MV designed the experiments. CMP performed immunostainings, super-resolution and release probability experiments and analyzed the data. AM performed axonal/dendritic DCV fusion experiment. JPN performed *in vivo* DCV labeling. MF designed and performed the initial colocalization and release probability experiments. AM, JHB, SA, and ND performed glutamatergic/GABAergic release probability experiment. JRTW collected and analyzed electron microscopy data. CMP, RFT, and MV designed figures and wrote the manuscript with input from all authors.

## Conflict of interest

The authors declare that they have no conflict of interest.

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
