## [Review Process File · The EMBO Journal]

Pool size estimations for dense-core vesicles in mammalian CNS neurons

Claudia M. Persoon, Alessandro Moro, Joris Nassal, Margherita Farina, Jurjen H. Broeke, Swati Arora, Natalia Dominguez, Jan R.T. van Weering, Ruud F.G. Toonen, Matthijs Verhage

Review timeline:

Submission date:	19th Apr 18
Editorial Decision:	28th May 18
Revision received:	11th Jul 18
Editorial Decision:	7th Aug 18
Revision received:	8th Aug 18
Accepted:	8th Aug 18

Editor: Karin Dumstrei

Transaction Report:

1st Editorial Decision

28th May 18

Thank you for submitting your manuscript to The EMBO Journal. Your study has now been seen by two referees and their comments are provided below.

As you can see from the comments, the referees find the analysis insightful and important. They raise a number of different concerns that I anticipate you should be able to sort out in a good way. Given the feedback from the referees I would like to invite you to submit a suitably revised manuscript that addresses the raised concerns.

When preparing your letter of response to the referees' comments, please bear in mind that this will form part of the Review Process File, and will therefore be available online to the community. For more details on our Transparent Editorial Process, please visit our website:
http://emboj.embopress.org/about#Transparent_Process

Thank you for the opportunity to consider your work for publication. I look forward to your revision.

REFeree COMMENTS

Referee #1:

In this study, Verhage and colleagues carried out a thorough and quantitative analysis of large dense-core vesicles, distinguished by the presence of chromogranin B, in striatal and hippocampal neurons. While cultured neurons were used in most of the experiments, some additional experiments were also performed *in situ*, yielding similar numbers. The study combines modern microscopic techniques such as dSTORM and electron microscopy to carry out the first (as far as I know) quantification of the LDV inventory of CNS neurons.

While many of the data confirm previous studies (particularly with respect to the stimulation of LDV exocytosis), this study stands out as it is comprehensive and scholarly conducted, filling in quantitative details that were missing in the field since many years. Thus, I am supportive of publication provided that my concerns about the differentiation between dendritic, axonal, and synaptic LDVs are addressed as outlined below.

Major point:

To me it is not clear how well axons and dendrites were separated in the DCV counting experiments (as shown in Fig. 3). It is well known that axons run along dendrites (particularly in microisland cultures!) where they form varicosities (as acknowledged by the authors: Fig. S1), making it very difficult to distinguish dendritic LDVs from those contributed by such accompanying axons. Thus, the data shown for single neurons in Fig. 3 are probably compromised, and I am not sure this is much better in the network cultures. In fact, I find it surprising that the authors did not use a bona-fide axonal marker for better differentiating between axons and dendrites instead of the pan-neurite marker β 3-tubulin that fails to identify dendrite-accompanying axons. This is a critical issue since the differential quantification of LCVs in axons and dendrites, respectively, is one of the main results of the manuscript. The same problem affects the light microscopy analysis of DCVs in synapses: In my opinion, it cannot be excluded that the DCVs assigned to presynaptic terminals (identified by VGLUT staining) are contributed by postsynaptic DCVs that cannot be resolved. Clarity is only obtained by the EM analysis. Similarly, the assignment of exocytotic events to axons and dendrites is problematic as axons and dendrites were only identified by their morphological features. In my opinion, more stringent criteria are needed to support the assignment of LDVs to axons and dendrites.

Minor points:

1. The figure legends are terse, and in several cases it is difficult to understand details that are hidden in the methods part or (at least as far as I have seen) are entirely missing. For example, at the bottom of the multipanel Fig. 1 it is stated that "N numbers represent number of independent experiments and individual observations in brackets". What does this mean? Are the numbers in brackets referring to individual neurons, neurites, or else in the different panels? Are the statistical calculations based on independent experiments or on "individual observations", whatever they are? Please explain.
2. Fig. 2a is unclear: Is the green signal based on dSTORM or is this confocal resolution? What are the curves and arrows signifying? In the left panel: Are the colors shown here only for clarification or based on data? How do the authors know which is a dendrite and which is the axon - based on neurite thickness?
3. While the localization of LDVs with respect to SV clusters is interesting (Fig. 4), the distance analyses need to be taken with a grain of salt. In particular, the distance of LDVs to SV clusters are based on the arbitrary positioning of the section and thus represent upper limits - they may in fact be closer when considering the 3D structure of the terminal.

Referee #2:

In this study, the Authors analyze the dense core vesicle (DCV) pool using a combination of techniques including confocal-, electron- and super-resolution microscopy. They quantify DCV amounts in excitatory hippocampal and inhibitory striatal neurons *in vitro* and in thalamic cortical axons *in vivo*, providing information about their localization in axons and dendrites. They conclude that the DCV pool ranges from 1,400 to 18,000 DCVs per neuron and correlates with axon length. They also find that the DCV releasable pool is about 20-400 vesicles, depending on type of stimulation, and that fusion events occur mostly in the axon.

These results may be of interest for the scientific community, as they provide important information about key properties of DCVs, which are in general much less characterized than typical synaptic vesicles. A limit of the study -or at least of the first part of it- is that, although having available high resolution techniques such as electron microscopy and dSTORM, the Authors draw conclusions based on the results of confocal analysis (which lacks the necessary resolution) performed in unsuited cell models (e.g. isolated neurons growing in culture). Although the Authors seem to be aware of these limits (lines 106-107, line 148), still a relevant part of their conclusions are based on these data (see for example Fig. 1 and Fig. 3).

More specifically:

In Fig. 1 the Authors quantify the number of DCVs in neuronal processes using antibodies against ChgA/ChgB and confocal analysis. This is rather questionable, as confocal microscopy does not allow to discriminate single DCVs. To resolve this problem (lines 106-107), the Authors use dSTORM imaging (Fig. 2), which is however performed only in hippocampal neurons. The lack of corresponding data for inhibitory striatal neurons *in vitro* and thalamic cortical axons *in vivo* makes the quantitative comparisons reported in fig. 1 not completely reliable.

The confocal analysis of Fig. 3 lacks the resolution necessary to univocally determine the axonal versus dendritic distribution of DCVs. In primary cultures -especially in neurons growing in isolation- dendrites and axons frequently grow close to each other. Also, ChgB positive puncta appear frequently as leaning on processes (see Fig 3C and D), making questionable their true localization. The Authors acknowledge this problem (line 148) and perform a minor part of their analysis in sparse cultures (Fig. 3N and O), still by confocal analysis and without providing any representative picture. This reviewer wonders why the Authors do not take advantage of the advanced techniques they have available (EM and dSTORM) to reliably quantify dendritic versus axonal DCVs in suited experimental models.

The localization of DCVs in neuronal processes *in vivo* is a key issue which has never been fully addressed before. The Authors investigate this point in cultured neurons (Fig. 3), with the limits described above. It would be very relevant if the Authors could provide similar data *in vivo*. In Fig. 2, the Authors only analyze thalamo-cortical axonal projections. Could they provide *in vivo* information about DCV localization in axons versus dendrites, for example at the hippocampal level?

Minor issues:

Based on the fact that confocal analysis does not allow to discriminate single DCVs, the Authors should avoid saying that they use this method "to quantify the total number of DCVs per neuron" (line 89).

Fig. 2 A: please explain how the identification of the axon and the dendrite was performed. In some pictures it is clear that the dSTORM and the confocal staining for ChgB completely lack correspondence (see for example the right end of the examined process in fig. 2D). Why is that?

Lines 291-296 in Discussion could probably be moved to Results.

When discussing release of DCVs upon high frequency stimulation, the Authors may wish to quote the pioneer studies of Andersson and colleagues (J Physiol. 1982) and the historical review of Lundberg and Hokfelt (TINS, 1983).

We thank the reviewers for their insightful comments and constructive suggestions. To address the reviewers' comments, we have added 5 new data sets with analysis of DCV distribution in axons and dendrites of hippocampal neurons *in vivo*, better analysis of axonal versus dendritic distribution with the bona-fide axon marker SMI312, and high resolution dSTORM imaging in striatal neurons and STED imaging in thalamo-cortical axons to fully comply with the reviewers' suggestions (Figures 2E-I, 3N-O, 3R-S, expanded view figure 2). Together, we feel this has considerably strengthened the conclusions of the data.

Reviewer #1:

This reviewer raises 1 major issue and mentions 3 minor issues.

*In this study, Verhage and colleagues carried out a thorough and quantitative analysis of large dense-core vesicles, distinguished by the presence of chromogranin B, in striatal and hippocampal neurons. While cultured neurons were used in most of the experiments, some additional experiments were also performed *in situ*, yielding similar numbers. The study combines modern microscopic techniques such as dSTORM and electron microscopy to carry out the first (as far as I know) quantification of the LDV inventory of CNS neurons. While many of the data confirm previous studies (particularly with respect to the stimulation of LDV exocytosis), this study stands out as it is comprehensive and scholarly conducted, filling in quantitative details that were missing in the field since many years. Thus, I am supportive of publication provided that my concerns about the differentiation between dendritic, axonal, and synaptic LDVs are addressed as outlined below.*

We thank the reviewer for his/her positive evaluation of our manuscript.

Major issue:

To me it is not clear how well axons and dendrites were separated in the DCV counting experiments (as shown in Fig. 3). It is well known that axons run along dendrites (particularly in microisland cultures!) where they form varicosities (as acknowledged by the authors: Fig. S1), making it very difficult to distinguish dendritic LDVs from those contributed by such accompanying axons. Thus, the data shown for single neurons in Fig. 3 are probably compromised, and I am not sure this is much better in the network cultures. In fact, I find it surprising that the authors did not use a bona-fide axonal marker for better differentiating between axons and dendrites instead of the pan-neurite marker β 3-tubulin that fails to identify dendrite-accompanying axons. This is a critical issue since the differential quantification of LCVs in axons and dendrites, respectively, is one of the main results of the manuscript.

Reply: We fully agree with the reviewer that this is an important issue. We have repeated our initial experiment (Fig. 3H-M) in single neurons using the bona-fide axonal marker SMI312 and dendritic marker MAP2. This new dataset (new Fig. 3N-O) shows a similar distribution and intensity of ChgB puncta in dendrites and axons compared to the initial dataset; ChgB puncta are more densely packed in dendrites with a higher intensity per puncta compared to axonal ChgB puncta. As the total dendritic length is much smaller than axonal length, we conclude that the distribution of DCVs is similar between axons and dendrites. Furthermore, upon suggestion of reviewer #2 (see below) we quantified the distribution of DCVs in dendrites and axons *in vivo* (Fig. 3R-S), which shows a similar distribution of ChgA puncta in dendrites and axons of dentate gyrus granule cells, in line with our *in vitro* analysis.

The same problem affects the light microscopy analysis of DCVs in synapses: In my opinion, it cannot be excluded that the DCVs assigned to presynaptic terminals (identified by VGLUT staining) are contributed by postsynaptic DCVs that cannot be resolved. Clarity is only obtained by the EM analysis.

Reply: Confocal microscopy resolution indeed does not allow discrimination between pre- and postsynaptic regions. Therefore, we constrain our conclusions in figure 3G as “synaptic” or “extra-synaptic” using vGLUT1 as synaptic marker and do not make the distinction between pre- or post-

synaptic. This also holds for our dSTORM analysis (Fig. 4A-C) where the VGLUT1 signal is represented at confocal resolution. However, our EM analysis showed a clear localization of DCVs to presynaptic compartments. In 110 random synaptic sections, only 5 showed a DCV in the postsynaptic compartment. We clarified this issue in the revised manuscript (p9, 1188, 190-191) We therefore conclude based on EM data that the vast majority of DCVs are present at the periphery of the pre-synaptic vesicle cluster.

Similarly, the assignment of exocytotic events to axons and dendrites is problematic as axons and dendrites were only identified by their morphological features. In my opinion, more stringent criteria are needed to support the assignment of LDVs to axons and dendrites.

Reply: We agree with the reviewer (see above) and performed post-hoc immunostainings using the axonal marker SMI312 and dendritic MAP2 to better distinguish between axonal and dendritic release. New Expanded View Fig. 2 together with figure 6 shows that DCVs preferentially fuse in SMI312 labeled axons, in line with our initial observations.

Minor points:

1. The figure legends are terse, and in several cases it is difficult to understand details that are hidden in the methods part or (at least as far as I have seen) are entirely missing. For example, at the bottom of the multipanel Fig. 1 it is stated that "N numbers represent number of independent experiments and individual observations in brackets". What does this mean? Are the numbers in brackets referring to individual neurons, neurites, or else in the different panels? Are the statistical calculations based on independent experiments or on "individual observations", whatever they are? Please explain.

Reply: We apologize that the figure legends and methods were not always clear and have added better descriptions and explanations. The numbers in brackets refer to individual neurons or individual measurements when applicable. The N numbers represent individual experiments. Statistical calculations are based on individual neuron measurements.

2. Fig. 2a is unclear: Is the green signal based on dSTORM or is this confocal resolution? What are the curves and arrows signifying? In the left panel: Are the colors shown here only for clarification or based on data? How do the authors know which is a dendrite and which is the axon - based on neurite thickness?

Reply: We apologize the figure was unclear. We have adjusted the figure and extended the legend to clarify. The green signal represents the confocal signal and the curves represent the point-spread function. The colors are all shown for clarification. In the EM figure we based our conclusion of the type of neurite on neurite thickness.

3. While the localization of LDVs with respect to SV clusters is interesting (Fig. 4), the distance analyses need to be taken with a grain of salt. In particular, the distance of LDVs to SV clusters are based on the arbitrary positioning of the section and thus represent upper limits - they may in fact be closer when considering the 3D structure of the terminal.

Reply: We agree completely with the reviewer. In 2D analyses, we cannot exclude that outside the plain of the 2D section, another part of the plasma membrane and/or the active zone is in fact closer to the DCV. We have clarified this point in the revised version of the manuscript (p9, 1199-200).

Reviewer #2

This reviewer raises 3 major issues and 4 minor issues.

In this study, the Authors analyze the dense core vesicle (DCV) pool using a combination of techniques including confocal-, electron- and super-resolution microscopy. They quantify DCV amounts in excitatory hippocampal and inhibitory striatal neurons in vitro and in thalamic cortical axons in vivo, providing information about their localization in axons and dendrites. They conclude that the DCV pool ranges from 1,400 to 18,000 DCVs per neuron and correlates with axon length.

They also find that the DCV releasable pool is about 20-400 vesicles, depending on type of stimulation, and that fusion events occur mostly in the axon. These results may be of interest for the scientific community, as they provide important information about key properties of DCVs, which are in general much less characterized than typical synaptic vesicles.

We thank the reviewer for his/her positive evaluation of our manuscript.

1- A limit of the study -or at least of the first part of it- is that, although having available high resolution techniques such as electron microscopy and dSTORM, the Authors draw conclusions based on the results of confocal analysis (which lacks the necessary resolution) performed in unsuited cell models (e.g. isolated neurons growing in culture). Although the Authors seem to be aware of these limits (lines 106-107, line 148), still a relevant part of their conclusions are based on these data (see for example Fig. 1 and Fig. 3). More specifically: In Fig. 1 the Authors quantify the number of DCVs in neuronal processes using antibodies against ChgA/ChgB and confocal analysis. This is rather questionable, as confocal microscopy does not allow to discriminate single DCVs. To resolve this problem (lines 106-107), the Authors use dSTORM imaging (Fig. 2), which is however performed only in hippocampal neurons. The lack of corresponding data for inhibitory striatal neurons in vitro and thalamic cortical axons in vivo makes the quantitative comparisons reported in fig. 1 not completely reliable.

Reply: This is a valid point. To address this, we have now performed dSTORM imaging also on cultured striatal neurons (new Fig. 2E-G). In addition, we have now also performed high resolution STED microscopy on thalamo-cortical axons *in vivo* (new Fig. 2H-I). These new data corroborate our previous conclusions on ChgB puncta in hippocampal neurons and show that the average 1360 ChgB puncta in striatal neurons observed in confocal microscopy (Fig. 1H) represent a total DCV pool of approximately 3730 DCVs per neuron. Furthermore, the average of 0.53 ChgA puncta per μm axon (Fig. 1P) represent approximately 0.72 DCVs per μm axon *in vivo*. These new and more exact estimations have been added to the revised manuscript (p6-7, 1123-125, 127-135)

2- The confocal analysis of Fig. 3 lacks the resolution necessary to univocally determine the axonal versus dendritic distribution of DCVs. In primary cultures -especially in neurons growing in isolation- dendrites and axons frequently grow close to each other. Also, ChgB positive puncta appear frequently as leaning on processes (see Fig 3C and D), making questionable their true localization. The Authors acknowledge this problem (line 148) and perform a minor part of their analysis in sparse cultures (Fig. 3N and O), still by confocal analysis and without providing any representative picture. This reviewer wonders why the Authors do not take advantage of the advanced techniques they have available (EM and dSTORM) to reliably quantify dendritic versus axonal DCVs in suited experimental models.

Reply: We agree that we could have taken more advantage of the advanced techniques we have available, although for DCV pool size estimations, it remains inevitable to make translations between confocal microscopy and more advanced techniques (a representative number of whole neurons cannot realistically be obtained using these high resolution techniques). In any case, we have now performed several new experiments using dSTORM and also STED, also on *in vivo* neurons (see point #1 above and #3 below). Concerning 'axonal versus dendritic distribution of DCVs', we now added new data to Fig 3 using the bona-fide axonal marker SMI312 to better distinguish between axon and dendrites. This new data set (new Fig. 3N-O) shows that ChgB puncta are more densely packed in dendrites with higher intensity per puncta compared to axonal ChgB puncta, similar to our initial observations. As dendrites are much shorter than axons, we conclude that the distribution of DCVs is similar between axons and dendrites.

3- The localization of DCVs in neuronal processes in vivo is a key issue which has never been fully addressed before. The Authors investigate this point in cultured neurons (Fig. 3), with the limits described above. It would be very relevant if the Authors could provide similar data in vivo. In Fig. 2, the Authors only analyze thalamo-cortical axonal projections. Could they provide in vivo information about DCV localization in axons versus dendrites, for example at the hippocampal level?

Reply: We agree that analysis of DCV localization *in vivo* is a key issue of our manuscript. To address the reviewer's question, we analyzed DCV distribution in hippocampal dentate gyrus

granule cells by injecting mCherry AAV (similar as described in figure 1L). Slices were obtained using the “magic cut” to ensure visualization DG granule cell axons and dendrites (Bischofberger *et al*, 2006). We quantified the distribution of ChgA labeled DCVs in dendrites and axons of these neurons (new Fig. 3R,S). This showed a similar distribution of DCV puncta per μm axon and dendrite, in line with our analysis in cultured neurons. We have added this important new conclusion to the revised version of the manuscript (p8, 1174-180)

Minor issues:

Based on the fact that confocal analysis does not allow to discriminate single DCVs, the Authors should avoid saying that they use this method "to quantify the total number of DCVs per neuron" (line 89).

Reply: We agree and have changed this sentence to “to estimate the total number of DCVs per neuron”.

Fig. 2 A: please explain how the identification of the axon and the dendrite was performed. In some pictures it is clear that the dSTORM and the confocal staining for ChgB completely lack correspondence (see for example the right end of the examined process in fig. 2D). Why is that?

Reply: We have identified the axon and dendrite in fig. 2A based on morphology. Also based on suggestion of reviewer 1 we have now adjusted figure 2A for better clarification and have extended the figure legend. The lack of correspondence between dSTORM and confocal staining for ChgB can be explained by the presence of the ChgB signal in another z-plane. Confocal imaging of ChgB using a 40x magnification captures a wider z-plane compared to dSTORM imaging, which was performed using a 100x magnification and in oblique TIRF.

Lines 291-296 in Discussion could probably be moved to Results.

Reply: We agree and have moved these sentences to the results (p.10, 1208-211) as suggested.

When discussing release of DCVs upon high frequency stimulation, the Authors may wish to quote the pioneer studies of Andersson and colleagues (J Physiol. 1982) and the historical review of Lundberg and Hokfelt (TINS, 1983).

Reply: We thank the reviewer for pointing out this omission. We have added these seminal papers to the discussion (p.14,15, 1317-320).

Other changes

One of the co-authors felt in hind side that she didn't contribute enough to the manuscript to award authorship.

References:

Bischofberger J, Engel D, Li L, Geiger JRP & Jonas P (2006) Patch-clamp recording from mossy fiber terminals in hippocampal slices. *Nat. Protoc.* **1**: 2075–2081

2nd Editorial Decision

7th Aug 18

Dear Matthijs,

Thanks for sending the revised manuscript. Your study has now been seen by the two referees and their comments are provided below. As you can see both referees appreciate the introduced changes and support publication here.

Just a few minor changes are needed (see referee #2's comments) before formal acceptance here.

When you re-submit the revised version will you also take care of the following items:

- Callout to Figure Fig. 4 A & B are missing.
- There is a callout to Figure S2 on page 13 - did you mean Figure EV2?
- Our publisher Wiley has done their pre-publication check on your manuscript. I have attached their corrected version - please see figure legends. Please take a look at their comments and incorporate their suggestions.

That should be all. Once we get the revised version in I will send you the acceptance letter.

REFEREE COMMENTS

Referee #1:

During revision, the authors have added new evidence, particularly with respect to differentiating between axons and dendrites (using MAP-2 staining). These data have considerably strengthened the manuscript, and the major conclusions are, in my view, now fully supported by the data that are of very high quality. For these reasons, I recommend acceptance of the revised manuscript.

Referee #2:

I read and compared the two versions of the manuscript, considering the requests made by Reviewers. I think that the authors successfully addressed all the raised points. They incorporated the required new data to figures 2 and 3 and introduced new additional figures.

My only suggestion for the Authors is to include the staining for ChGA in figure 3, which reports the novel experiments aimed at quantifying the DCV content in the hippocampus. While both channels were properly displayed in Fig. 1 (mCherry-filler and Alexa Fluor antibody-ChGA), the latter is missing in Fig. 3R, despite the figure legend mentions "Representative mouse brain slice with labeled DG granule cells (mCherry-filler, red) and immunostained for ChgA (green)". Given ChGA is the object of quantification.

Corresponding Author Name: Matthijs Verhage

Journal Submitted to: The EMBO Journal

Manuscript Number: EMBOJ-2018-99672